# Aftershock sequences and seismic-like organization of acoustic events produced by a single propagating crack

Jonathan Barés[1,2], Alizée Dubois[1], Lamine Hattali[1,3], Davy Dalmas[4] & Daniel Bonamy [1]

Brittle fractures of inhomogeneous materials like rocks, concrete, or ceramics are of two types: Nominally brittle and driven by the propagation of a single dominant crack or quasi-brittle and resulting from the accumulation of many microcracks. The latter goes along with acoustic noise, whose analysis has revealed that events form aftershock sequences obeying characteristic laws reminiscent of those in seismology. Yet, their origin lacks explanation. Here we show that such a statistical organization is not only specific to the multi-cracking situations of quasi-brittle failure and seismology, but also rules the acoustic events produced by a propagating crack. This simpler situation has permitted us to relate these laws to the overall scale-free distribution of inter-event time and energy and to uncover their selection by the crack speed. These results provide a comprehensive picture of how acoustic events are organized upon material failure in the most fundamental of fracture states: single propagating cracks.

[1] Service de Physique de l'Etat Condensé, CEA, CNRS, Université Paris-Saclay, CEA Saclay, 91191 Gif-sur-Yvette, Cedex, France. [2] Laboratoire de Mécanique et Génie Civil Université de Montpellier CNRS 163 rue Auguste Broussonnet, 34090 Montpellier, France. [3] Laboratoire FAST, Université Paris-Sud, CNRS Université Paris-Saclay F-91405 Orsay, France. [4] Laboratoire de Tribologie et Dynamique des Systemes, CNRS Ecole Centrale de Lyon 36, Avenue Guy de Collongue, 69134 Ecully, Cedex, France. Correspondence and requests for materials should be addressed to D.B. (email: daniel.bonamy@cea.fr)

Stress enhancement at defects makes the damage behavior observed at the continuum-level scale extremely dependent on material microstructure down to very-small scales. This results in large statistical fluctuations in the fracturing behavior at the macroscopic scale, which are difficult to control in practice. For homogeneous brittle solids under tension, the difficulty is tackled by reducing the problem down to that of the destabilization and further growth of a single pre-existing crack[1]. Linear elastic fracture mechanics then provides the relevant theoretical framework to describe crack propagation in homogeneous materials[1], and the use of some concepts coming from out-of-equilibrium physics permits a global self-consistent approach of crack propagation in the presence of weak heterogeneities[2]. The problem becomes a priori different in heterogeneous materials for loading conditions stabilizing crack propagation (such as compression). In these situations of so-called quasi-brittle failure, the material starts accumulating diffuse damage through barely perceptible microfracturing events; then it collapses abruptly when a macroscopic crack percolates throughout the microcrack cloud[3]. Quasi-brittle failure can also be promoted in specimens upon tension by a higher degree of heterogeneity in the material[4], lower strain rate[5], and more active chemical environments[6].

Today's most widely used technique to probe damage evolution in quasi-brittle fracture consists in monitoring acoustic emission. This provides a sensitive non-intrusive method to detect microfracturing events and localize them in both time (μs resolution) and space (coarser resolution). A geophysical-scale analogy is seismicity analysis in the mitigation of earthquake hazard. In both cases, acoustic events (AE) display similar scale-free dynamics organized into mainshock (MS)–aftershock (AS) sequences characterized by a range of empirical scaling laws: First stated by Omori in 1894[7], and refined later by Utsu[8], the AS frequency decays algebraically with time from MS. Next, the Gutenberg–Richter law asserted in 1944[9] that the event frequency decays as a power-law with energy (or equivalently the frequency decays exponentially with the event magnitude). In 1965, the Båth's law[10] affirmed that the difference in magnitude between a MS and its largest AS is constant, independent of the MS magnitude; the so-called AS productivity law[11,12] states that the number of produced AS increases as a power-law with the energy of the triggering MS. Most recently Bak et al. (2002)[13] showed that, once rescaled by the activity rate, the distribution of inter-event times obeys a unified scaling law. These laws are central in the implementation of probabilistic forecasting models for seismic hazard[14].

These laws have proven of general validity, in natural[13,15] and induced[16,17] seismicity at the geophysical scale, and in quasi-brittle fracture experiments at the lab scale, either upon compression[18–20] or caused by the release of a gas[21]. Yet, they remain empirical. They are usually seen as emergent properties for the collective dynamics of microcrack nucleation, structured by the long-range stress redistribution following each microfracturing event[22,23]. Still, the dynamics of a single peeling front propagating along a two-dimensional heterogeneous interface is governed by local and irregular jumps[24,25], the size, occurrence time, and occurrence location of which share statistical similarities with that of earthquakes[26]. What if the time organization of events find its origin in the simpler and more tractable problem of a unique nominally brittle crack propagating in an heterogeneous solid?

Here, we analyze the time–energy organization of AE that accompanies the slow stable propagation of a single brittle crack throughout an artificial rock made of sintered monodisperse polystyrene beads (Methods). In the homogeneous parent polymer specimen, such a crack propagates continuously and regularly and no AE occur. On the other hand, increasing the microstructure scale (the diameter, $d$, of the sintered beads) unveils irregular burst-like dynamics and numerous AE

accompanying the crack front's movement. As in the multi-cracking situations of quasi-brittle fracture, the events form MS-AS sequences obeying the fundamental scaling laws of statistical seismology: the Omori–Utsu law, the productivity law and Båth's law. Nonetheless, in this situation of single crack propagation, the above seismic laws are demonstrated to emerge directly from the scale-free statistics of energy (for the productivity law and Båth's law) and from that of inter-event time (for the Omori–Utsu law) according to relations that have been unraveled, without further information on time–energy correlations (or spatio-temporal correlations).

## Results

**Selection of the activity rate.** Figure 1a shows a typical time series of the AE observed for $d = 583$ μm and a mean crack speed $\bar{v} = 2.7$ μm s$^{-1}$. Note the variety of sizes, as evidenced by using the logarithmic scale. Eight transducers spatially localize the AE sources inside the specimen (Fig. 1b and Supplementary Movie 1). Within the localization resolution (~5 mm), the sources gather along the moving crack front (Fig. 1b, c, Supplementary Movie 1) as expected for nominally brittle fracture. This nominally brittle characteristics has also been demonstrated in earlier work from the proportionality between the elastic power released at each time step and the instantaneous crack speed[27]. In the present experiments, AE result from the local jumps of the front as it suddenly depins from heterogeneities, and not from the collective nucleation of microcracks spreading throughout the solid as in quasi-brittle failure situations.

The cumulative number of produced AE increases continuously and linearly with crack length. Moreover, the proportionality constant, $C$, is independent of mean crack speed, $\bar{v}$, over the region swept by the crack (Fig. 1d). This indicates that the mean number of AE produced as the crack propagates over a unit length is given by the number of heterogeneities met over this period: $C \approx H/d^2$, where $H$ is the specimen thickness. The measured values, $C = 53 \pm 3$ AE mm$^{-1}$ for $d = 583$ μm and $H = 15$ mm, and $C = 270$ AE mm$^{-1}$ for $d = 223$ μm and $H = 15$ mm, are in agreement with the values $C \approx 44$ heterogeneities mm$^{-1}$ and $C \approx 300$ heterogeneities mm$^{-1}$ expected from the preceding relation. As a result, the activity rate $R$ (defined as the mean number of AE produced per unit time) is given by $R \approx \bar{v}H/d^2$, where $\bar{v}$ is the mean crack speed (Supplementary Figs 1 and 2).

**The Gutenberg–Richter law and self-similarity.** We now turn to the global statistical characterization of the AE time series. In all the experiments, the probability density function, $P(E)$, decays as a power-law over nearly five decades up to an upper corner energy (Fig. 2a). It is well-fitted by:

$$P(E) \propto E^{-\beta}\exp(-E/E_0), \tag{1}$$

with $E \geq E_{min}$. The lower cutoff, $E_{min} = 10^{-4}$, is the same in all our experiments. It is set by the sensitivity of the acquisition system. Conversely, the exponent $\beta$ and the upper corner energy $E_0$ depend on both crack speed (slightly) and material microstructure (more importantly). We will return at the end of this section to the analysis of these dependencies. Equation 1 is reminiscent of the Gutenberg–Richter law. Note, however, that the energy distributions observed in seismology often take the form of a pure power-law. Then, earthquake sizes are more commonly quantified by their magnitude, which is linearly related to the logarithm of the energy[28]: $\log_{10}E = 1.5M + 11.8$. The energy distribution takes the classical Gutenberg–Richter frequency–magnitude relation: $\log_{10}N(M) = a - bM$, where $N(M)$ is the number of earthquakes per year with magnitude larger than $M$ and $a$ and $b$ are constants. The $b$-value relates to the exponent $\beta$ involved in Eq. 1 via: $\beta = b/1.5 + 1$.

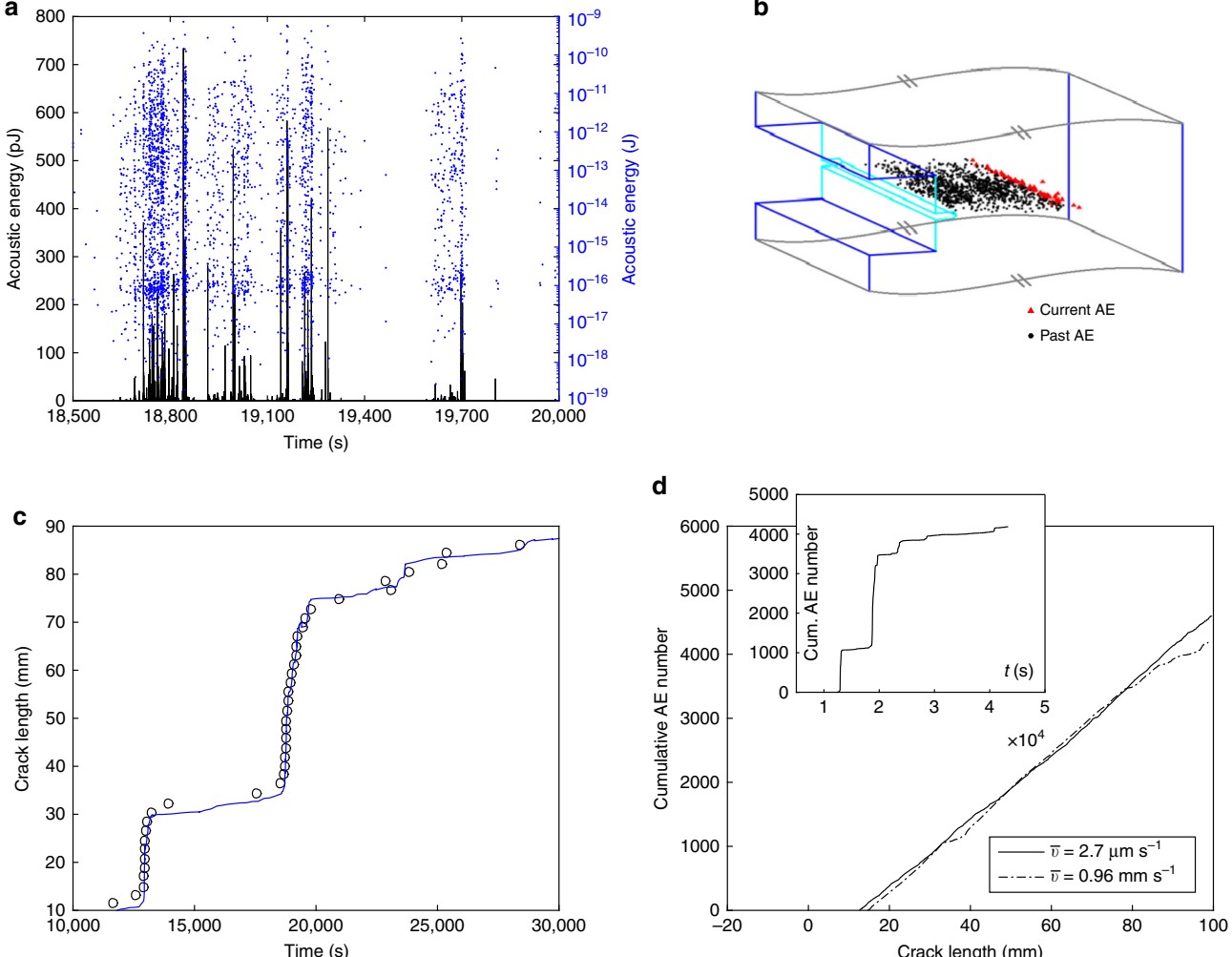

**Fig. 1** Acoustic emission going along with crack propagation in a inhomogeneous solid. **a** Typical snapshot showing the time evolution of the energy of recorded AE in linear (black) and logarithmic (blue) scales. The snapshot duration is 1500 s while the whole experiment lasts 5 h 30 minutes. **b** Schematic of the advancing crack front with localized AE (Supplementary Movie 1). Black points show all sources localized from the starting of the experiment. Red points only show AE emitted over the last second. AE gather in the immediate vicinity of the propagating front. **c** This point is confirmed by comparing the time evolution of the crack front as detected from a side-view imaging (blue line) with that of the AE center of mass over a moving time window of one second (open circles). **a**, **b**, and **c** all concern the same experiment, where the microstructure length-scale is $d = 583\,\mu m$ and the mean crack speed $\bar{v} = 2.7\,\mu m\,s^{-1}$. **d** main panel: Cumulative number of AE as a function of crack length in experiments with $\{d = 583\,\mu m, \bar{v} = 2.7\,\mu m\,s^{-1}\}$ (plain) and $\{d = 583\,\mu m, \bar{v} = 960\,\mu m\,s^{-1}\}$ (dash). **d** inset: Cumulative number of AE as a function of time for $\{d = 583\,\mu m, \bar{v} = 2.7\,\mu m\,s^{-1}\}$

Beyond the Gutenberg–Richter law, it has been demonstrated[13,15] that the recurrence times, $\Delta t$, of earthquakes with energies above a threshold value bound $E_{th}$ obey a unique universal distribution once time is rescaled with the rate of seismic activity over the considered energy range, $R(E_{th})$. Such a self-similar distribution is also observed in lab scale quasi-brittle fracture experiments[19–21]. The form of the rescaled distribution, $f$, depends on how the activity rate evolves with time[21]: For statistically stationary $R(t)$, $f(x)$ follows a gamma distribution[15,21] while, in the presence of a trend (that is a slowly varying component in the time series), $f(x)$ exhibits different power-law regimes[19,20]. Figure 2b shows $P(\Delta t)$ for different $E_{th}$ in a typical experiment and Fig. 2c shows the distribution after rescaling. The collapse and implied self-similarity are fulfilled. The scaled recurrence times obeys a gamma distribution:

$$P(\Delta t|E_{th}) = R(E_{th})f(u = \Delta t R(E_{th})),\qquad(2)$$

with $f(u) \propto u^{-\gamma}\exp(-u/B)$ for $u > b$. This underpins a stationary statistics for the AE series. This distribution involves three

parameters, which are interrelated (Supplementary Note 1 and Supplementary Eq. 6): The exponent $\gamma$ and the two rescaled time scales $b$ and $B$.

**Aftershock sequences and seismic laws.** The next step is to identify the AS sequences and to characterize their time–energy organization. In seismology, there exists powerful declustering methods to separate earthquakes into independent (background or MS) and dependent (offspring or AF) earthquakes[29]. Most of these methods are based on the spatio-temporal proximity of the events. Here, we adopted a procedure[19–21] used in compressive fracture experiments, where spatial information is not available and considered as MS all AE with energies in a predefined interval. The AS sequence following each of these MS is then defined as all subsequent AE, until an event of energy equal or larger than that of the MS is encountered.

Figure 3a shows the mean number of AS, $N_{AS}$, triggered by a MS of energy $E_{MS}$, in a typical fracture experiment. The

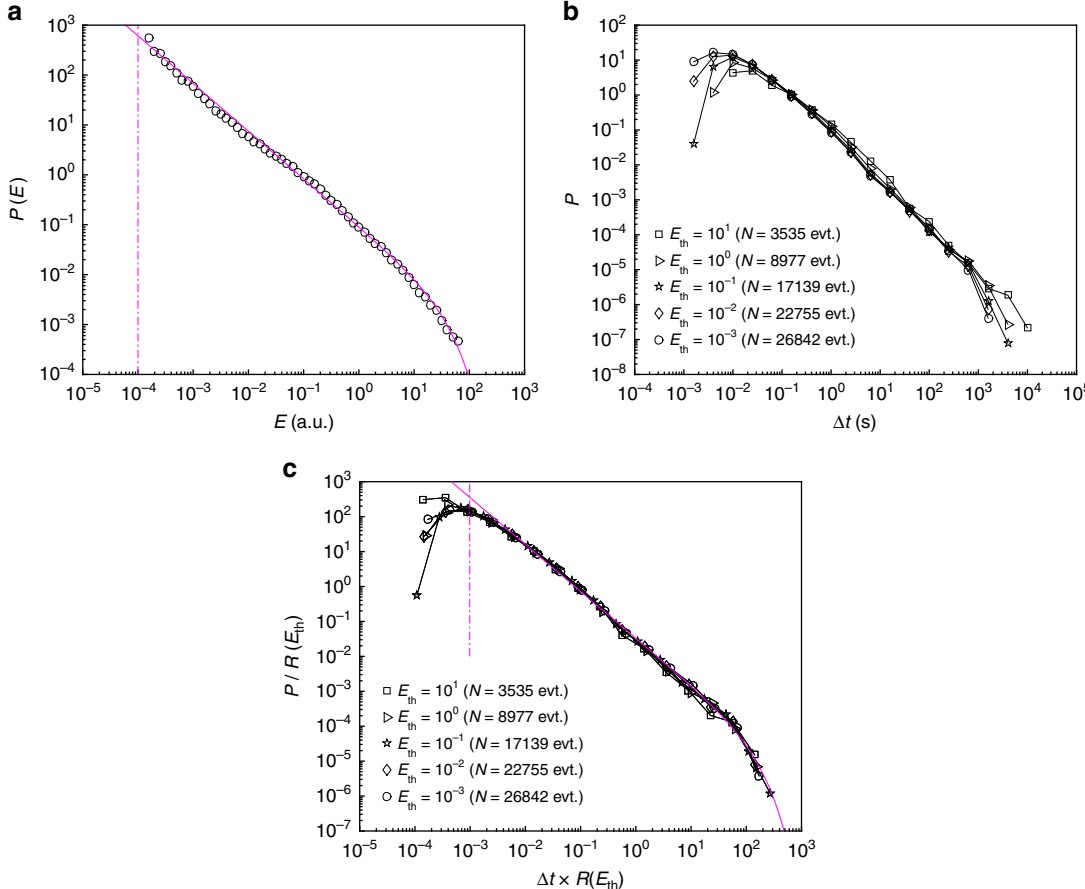

**Fig. 2** The Gutenberg–Richter law and time–energy self-similarity. **a** Distribution of AE energy in one of the experiments (microstructure length-scale: $d = 583\,\mu m$, crack speed: $\bar{v} = 2.7\,\mu m\,s^{-1}$). Solid magenta line is a gamma function $P(E) \propto E^{-\beta} exp(-E/E_0)$ for $E \geq E_{min} = 10^{-4}$ (vertical magenta dashed line), with fitted parameters $\beta = 0.96 \pm 0.02$ and $E_0 = 38 \pm 9$. **b** Distribution of the time interval $\Delta t$ separating two successive AE of energy larger than a prescribed energy threshold, $E_{th}$. **c** Collapse obtained after having rescaled $\Delta t$ with the mean activity rate $R(E_{th}) = N(E_{th})/T$, where $N(E_{th})$ is the number of AE with $E \geq E_{th}$ and $T = 31080\,s$ is the total duration of the fracture experiment. Magenta -curve is the gamma function $f(x) \propto x^{-\gamma} exp(-x/B)$ with $x \geq b$ with fitted parameters $\gamma = 1.34 \pm 0.03$, $B = 109 \pm 20$, and $b = 1 \pm 0.9 \times 10^{-3}$. $\pm$ stands for 95% confidence interval. In all panels, axes are logarithmic

productivity law is fulfilled, and $N_{AS}$ goes as a power-law with $E_{MS}$ as long as $E_{MS}$ is not too large (below a crossover energy value $E_c$). The curve remains unchanged after having permuted randomly the energy between the events (that is having attributed to each event $i$ the energy $E_j$ of another event $j$ chosen randomly), and having arbitrary set the time step to unity (that is having arbitrary set the time occurrence of the event $i$ to $t_i = i + 1$). This indicates that the productivity law, here, simply emerges from the Gutenberg–Richter distribution of the AE energy, without any further information on their time organization. Calling $F(E) = \int_{E_{min}}^{E} P(u)\mathrm{d}u$ the cumulative distribution of energy, we then expect (Supplementary Note 2):

$$N_{AS}(E_{MS}) = F(E_{MS})/(1 - F(E_{MS})), \qquad (3)$$

which compares very well with the experimental curve (Fig. 3a). When $\beta$ is larger than unity and the exponential cutoff in Eq. 1 can be neglected, this expression takes a simple scaling form (Supplementary Note 2): $N_{AS}(E_{MS}) \approx (E_{MS}/E_{min})^{\alpha}$ with $\alpha = \beta - 1$. Note that the measured curve $N_{AS}$ versus $E_{MS}$ a priori depends on the declustering method, that is on the algorithm used to decompose the catalog into AS sequences. It was checked that applying a different procedure does not affect significantly the form of this curve (Supplementary Fig. 3).

Båth's law states that the relative difference $\Delta M$ in magnitude ($M = \log_{10} E$) between the MS and its largest AS is constant,

independent of the MS energy. Figure 3b demonstrates this law is actually true here, as long as $E_{MS}$ is smaller than the crossover value $E_c$ defined from the productivity law. Above $E_c$, $\Delta M$ decays exponentially with $E_{MS}$. As for the productivity law, permuting randomly the events and setting arbitrary the time step to unity do not modify the curve. This implies that Båth's law finds its origin in the distribution of individual AE energy, without requiring further information on their overall sequencing. This picture is different from that provided in epidemic-type aftershock sequence (ETAS) models[14,30], where the series are built using a stochastic branching process and the Båth's law emerges from the correlations induced by the branching[31,32]. Here, extreme event theory permits to compute the statistics of the largest AS energy from the sequence triggered by a MS of prescribed energy, to compute its mean value, and finally to compute $\Delta M$ and its variations with $E_{MS}$ (Supplementary Note 3 and Supplementary Eq. 10). The predicted curve compares quite well with the experimental curve (Fig. 3b). As for productivity law, $\Delta M(E_{MS}/E_{min}|\beta)$ takes a simpler form when the energy distribution is a simple power law $P(E) \propto E^{-\beta}$ (Supplementary Note 3, Supplementary Eq. 12 and Supplementary Fig. 4).

We finally address the Omori–Utsu law, which states that the number of AS per unit time, $R_{AS}$, decays algebraically with the elapsed time since the MS occurrence, $t_{MS}$: $R_{AS}(t) = R_0/(1 + (t - t_{MS})/\tau)^p$, where $R_0$ and $\tau$ are characteristic rates and times, and $p$ defines the Omori exponent. In our experiments,

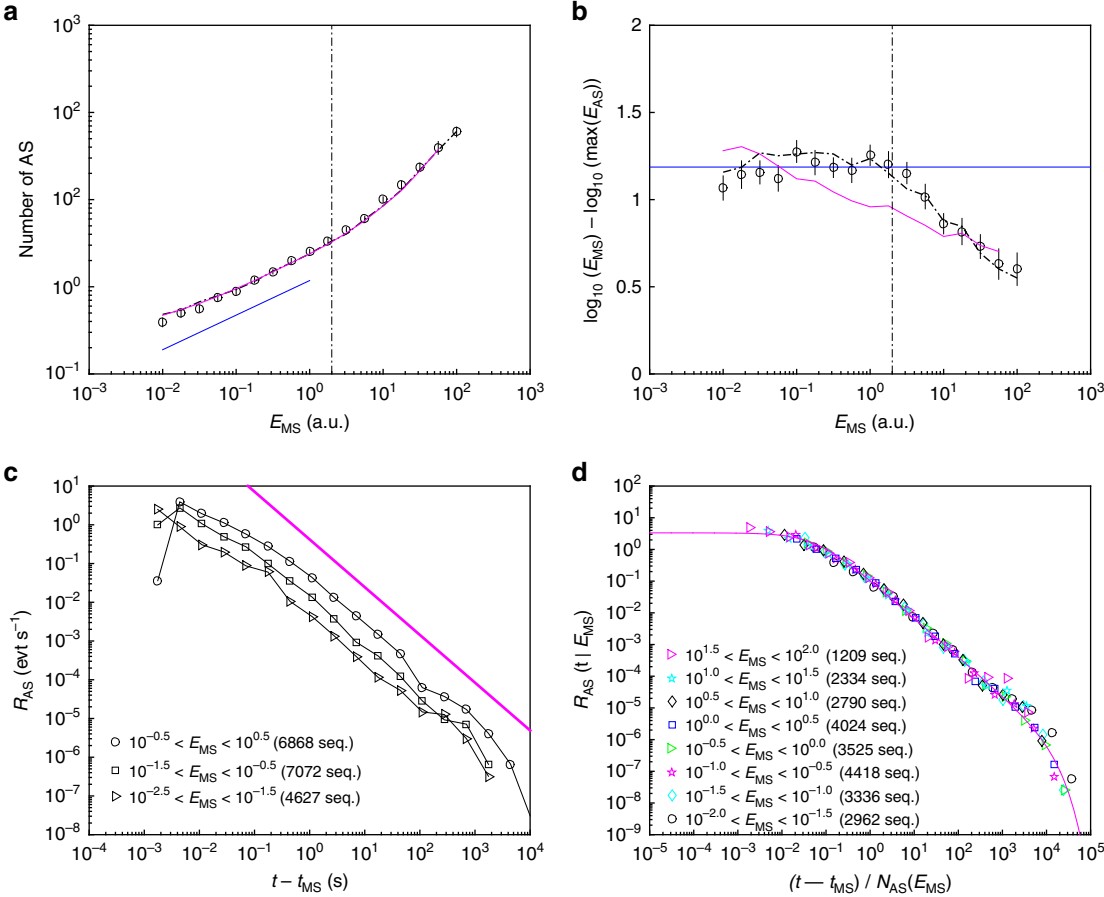

**Fig. 3** Aftershock sequences and seismic laws. All axes are logarithmic. **a** Circles show the variations of the mean AS number, $N_{AS}$, with the energy, $E_{MS}$, of the triggering MS. Dash-dotted line (barely visible beneath magenta line) shows the same curve after having permuted randomly all the events and arbitrary set the time step between two successive events to unity. Productivity power-law (straight blue line) is observed for $E_{MS} < E_c = 2$ (vertical dotted line). Magenta line is the curve predicted by Eq. 3 where the cumulative distribution $F(E)$ is the one measured in this experiment (integral of the curve presented in Fig. 2a). **b** Mean magnitude difference, $\Delta M = \log_{10} E_{MS} - \max(\log_{10} E_{AS})$, between the MS and its largest AS as a function of the MS energy. As in (a), circles show the experimental data while the dashed-dotted line shows that obtained after random permutation and setting time step arbitrary to unity. The error bars stand for 95% confidence interval. The plateau expected from Båth's law (horizontal blue line) is observed for $E_{MS} < E_c = 2$ (vertical dotted line). Its fitted value is $\Delta M = 1.19 \pm 0.05$. The magenta line is the curve predicted by Supplementary Equation 10. **c** Number of AS per unit time, $R_{AS}$, as a function of elapsed time since the MS occurrence, $t - t_{MS}$. Sequences have been sorted according to the MS energy as indicated in the legend. The magenta straight line indicates the Omori–Utsu power-law decay with a fitted exponent $p = 1.2 \pm 0.09$. **d** Rescaled Omori–Utsu plot $R_{AS}(t|R_{MS})$ as a function of $(t - t_{MS})/N_{AS}(E_{MS})$ where $N_{AS}(E_{MS})$ is given by Eq. 3. Note the perfect collapse of all curves, over all the accessible range in $E_{MS}$. The magenta line is a fit according to Eq. 4 with $p = 1.18 \pm 0.04$, $\tau_{min} = 0.06 \pm 0.02$ s, and $\tau_0 = 10 \pm 3$ ks. All panels concern the same experiment with $d = 583$ μm and $\bar{v} = 2.7$ μm s$^{-1}$

$R_{AS}(t|E_{MS})$ is computed by binning the AS events over $t - t_{MS}$, and subsequently averaging the so-obtained curves over all MS with energy falling into the prescribed interval. Figure 3c shows the so-obtained curves. The algebraic decay predicted by Omori is fulfilled, over almost five decades. The Omori–Utsu exponent $p$ is independent of $E_{MS}$. Conversely, the prefactor increases with $E_{MS}$.

As $N_{AS} = \int_{t_{MS}}^{\infty} R_{AS}(t|E_{MS}) \mathrm{d}t$, making the Omori–Utsu law consistent with the productivity law yields either $R_0$ or $\tau$ to be proportional to $N_{AS}(E_{MS})$. The former scaling proves to be wrong while the second yields a perfect collapse of the curves (Fig. 3d). The collapse also reveals an exponential cutoff in the Omori–Utsu law, which is finally written as:

$$R_{AS}(t|E_{MS}) = \frac{R_0}{\left(1 + \frac{t - t_{MS}}{\tau_{min} N_{AS}(E_{MS})}\right)^p} \exp\left(-\frac{t - t_{MS}}{\tau_0 N_{AS}(E_{MS})}\right) \quad (4)$$

The four constants, the Omori–Utsu exponent $p$, the lower and upper time scales $\tau_{min}$ and $\tau_0$, and the characteristic activity rate $R_0$ are interrelated (Supplementary Note 4 and Supplementary Eq. 17). The very same law holds for the foreshock (FS) rate $R_{FS}(t|E_{MS})$ versus time to MS, $t_{MS} - t$ (Supplementary Fig. 5). This symmetry along time reversal is a consequence from the fact that the AE time series, here, are stationary.

Note finally that permuting randomly the AE energy in the initial series does not modify the curve in Fig. 3d. Hence, the Omori–Utsu law and time dependency of $R_{AS}(t|E_{MS})$ do not emerge from correlations between time occurrence and energy, but simply results from the scale-free distribution $P(\Delta t)$; the dependency with $E_{MS}$, for its part, only intervenes in $R_{AS}(t|E_{MS})$ through $N_{AS}(E_{MS})$. As a consequence, the parameters at play in Eq. 4 relates to those in Eq. 2 (Supplementary Note 5, Supplementary Figs. 6 and 7, and Supplementary Eq. 18). The equivalence between the Omori–Utsu law and the scale-free distribution of $P(\Delta t)$ observed here differs from what is reported

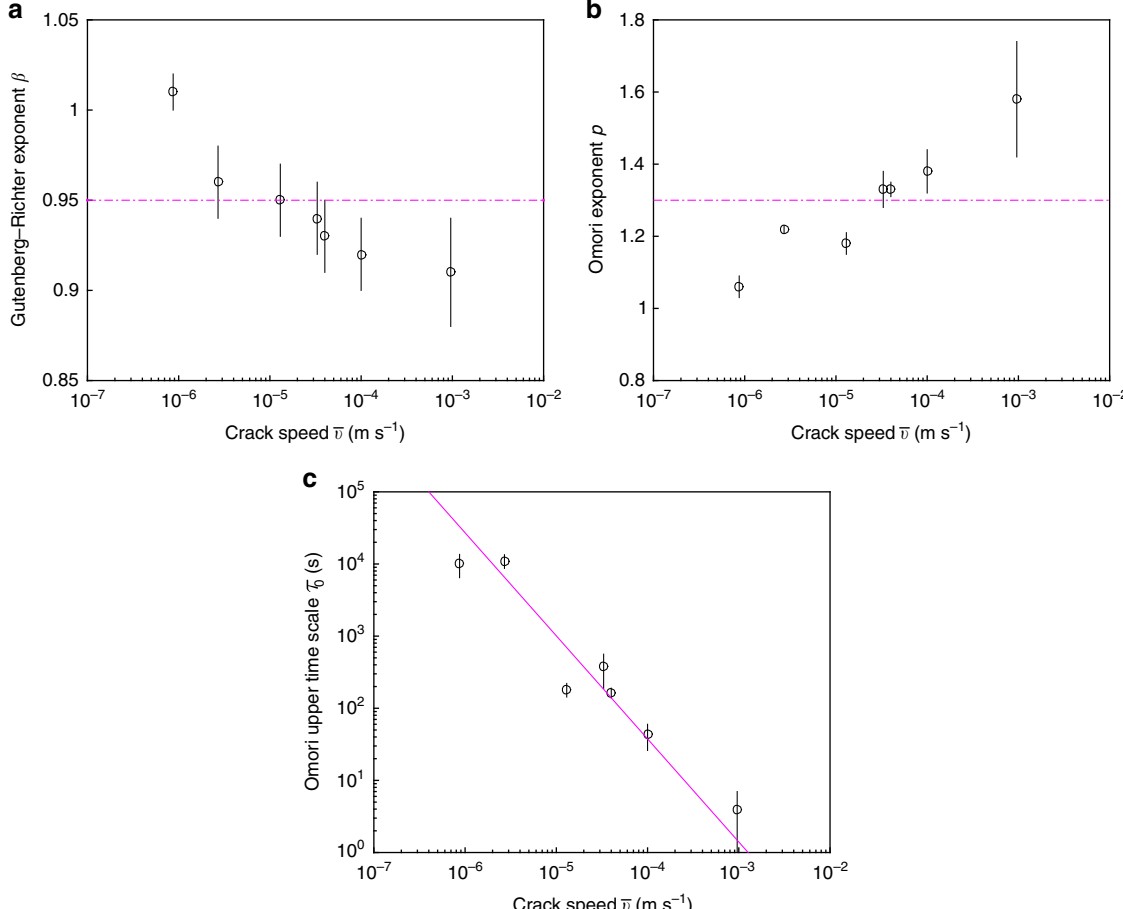

**Fig. 4** Effect of crack speed. **a** Variation of the Gutenberg–Richter exponent $\beta$ as a function of the mean crack speed $\bar{v}$. The horizontal magenta dash line indicates the mean value: $\bar{\beta} = 0.95$. **b** Variation of the Omori–Utsu exponent $p$ as a function of the $\bar{v}$. The horizontal magenta dashed line shows the mean value: $\bar{p} = 1.3$ **c** variation of the upper time scale $\tau_0$ associated with the rescaled Omori–Utsu law (Eq. 4). Straight magenta line is a power-law fit with exponent $-1/(2 - \bar{p})$. **a**, **b** x-axis is logarithmic. **c** both axes are logarithmic. In all panels, the errorbars stand for 95% confidence interval

in compressive fracture or seismicity (where the Omori–Utsu law implies the power-law distribution for inter-event times, but the reciprocal is not true[19,33]).

**Effect of crack speed**. The Gutenberg–Richter law, the unified scaling law for inter-event time, the productivity law, Båth's law and the Omori–Utsu law occur in all our experiments, irrespectively of the crack speed $\bar{v}$. Conversely the underlying parameters vary with $\bar{v}$. As demonstrated above, the productivity law and Båth's law are direct consequences of the Gutenberg–Richter distribution of energies, and the Omori–Utsu law for AS is equivalent to the power-law distribution of inter-event times. As a consequence, analyzing the effect of crack speed on the Gutenberg–Richter law and the Omori–Utsu law is sufficient to fully characterize its effect on the AE time–energy organization and the five associated seismic laws.

The lower cutoff for energy, $E_{\min} = 10^{-4}$, is independent of $\bar{v}$ and is set by the sensitivity of the acquisition system. The Gutenberg–Richter exponent $\beta$ logarithmically decreases with $\bar{v}$, from about 1 to 0.9 as $\bar{v}$ goes from $10^{-6}$ ms$^{-1}$ to $10^{-3}$ ms$^{-1}$ (Fig. 4a). This evolution can be due to the overlap of some AE, $\bar{v}$, and which increases with increasing $\bar{v}$, and which has been demonstrated[34] to lower the value of the effective exponent in systems containing temporal correlations. This may also be compared with other observations on quasi-brittle fracture experiments in rocks, which evidence a decrease of the

Gutenberg–Richter $b$-value (analog to $\beta$) with the loading rate[5] or stress intensity factor[6]. Note finally that, within the errorbars, the corner energy $E_0 \simeq 40$ does not evolve significantly with $\bar{v}$. The existence of such a corner energy for the Gutenberg–Richter law might find its origin in the finite size and/or the limited volume of material. Still, changing the microstructure length-scale $d$ while keeping the specimen dimensions constant also significantly affects both $E_0$ and the form of the cutoff function (Supplementary Fig. 8). The way the acoustic waves attenuate within the material (which depends on $d$) might then be a parameter to consider here.

Concerning the Omori–Utsu law (Eq. 4), increasing $\bar{v}$ yields a significant logarithmic increase of the exponent $p$, from about 1.1 to 1.6 as $\bar{v}$ goes from $10^{-6}$ ms$^{-1}$ to $10^{-3}$ ms$^{-1}$ (Fig. 4b). Conversely, it does not affect the characteristic time $\tau_{\min} \sim 0.05$ s. The latter closely resembles to the duration of the largest AE. The curve saturation observed for $t - t_{\rm MS} \leq \tau_{\min}$ is interpreted as the consequence of missing AS in the catalog right after the MS; their waveform having been drown in that of the MS. This mechanism is analog to the problem of short time aftershock incompleteness (STAI) documented in seismology[35,36] and responsible to some bias in the estimation of $\tau_{\min}$ (generally referred to as the $c$-value in the seismicity context). Finally, the upper corner time scale $\tau_0$ significantly decays with $\bar{v}$. This decrease can be predicted (Supplementary Note 6 and Supplementary Eq. 22), since $\tau_0$ is set by the upper time scale of the inter-event distribution and the activity rate is set by the crack speed $R \approx \bar{v}H/d^2$. Neglecting the

**Table 1 Synthesis of the samples and experiments analyzed here**

| Experiment No | Microstructure scale $d$ | Total AE number | Activity rate $R$ | Mean crack speed $\bar{v}$ |
|---|---|---|---|---|
| 1 | 583 μm | 33481 evt | $1.35 \times 10^{-1}$ evt s$^{-1}$ | 2.7 μm s$^{-1}$ |
| 2 | 583 μm | 5704 evt | $3.06 \times 10^{-2}$ evt s$^{-1}$ | 0.87 μm s$^{-1}$ |
| 3 | 583 μm | 18228 evt | 1.90 evt s$^{-1}$ | 33 μm s$^{-1}$ |
| 4 | 583 μm | 6063 evt | $6.78 \times 10^{-1}$ evt s$^{-1}$ | 13 μm s$^{-1}$ |
| 5 | 583 μm | 36795 evt | 2.10 evt s$^{-1}$ | 40 μm s$^{-1}$ |
| 6 | 583 μm | 31149 evt | 5.41 evt s$^{-1}$ | 100 μm s$^{-1}$ |
| 7 | 583 μm | 9133 evt | 22.8 evt s$^{-1}$ | 0.96 mm s$^{-1}$ |
| 8 | 233 μm | 160145 evt | 1.75 evt s$^{-1}$ | — |
| 9 | 233 μm | 65436 evt | 62.1 evt s$^{-1}$ | — |
| 10 | 24 μm | 21590 evt | 3.5 evt s$^{-1}$ | — |
| 11 | 24 μm | 19442 evt | 31.4 evt s$^{-1}$ | — |

Figures 1a, b, c, 2, 3 and Supplementary Figs. 2, 3, 4, 9, 10 and 11 involve experiment No. 1. Figure 1d involves experiments No. 1 and 7. Figure 4 and Supplementary Fig. 1 involve experiments No. 1 to 7. Supplementary Fig. 8 involves experiments No. 8 to 11

slight increase of $p$ with $\bar{v}$, this yields $\tau_0 \propto 1/\bar{v}^{1/(2-p)}$, which is compatible with the observations (Magenta line in Fig. 4c).

The above values correspond to materials with a microstructure length-scale $d = 583$ μm. It was checked that changing (reducing) $d$ does not change the picture: Scale-free statistics for energy and inter-event time, together with aftershock sequences obeying the productivity law, Båth's law and the Omori–Utsu law remain true regardless of the value of $d$. Conversely, the value of the exponents and the form of the cutoff function are material dependent and significantly evolve with specimen microstructure (Supplementary Fig. 8).

## Discussion
We have characterized here the statistical organization of the AE produced by a single crack propagating in a brittle heterogeneous material. The events form MS-AS sequences obeying the fundamental scaling laws of statistical seismology: The productivity law relating the AS number with the MS energy, Båth's law relating the magnitude of the largest AS with that of the MS, and the Omori–Utsu law relating the AS frequency with the time elapsed since MS. Hence, these laws are not specific to the multicracking situations of compressive quasi-brittle fracture or seismology, but extend to the far simpler situation of a single, slowly propagating, opening crack. In the latter case, they are direct consequences of the individual scale-free statistics of the energies (for the productivity law and Båth's law) and of inter-event times (for the Omori–Utsu law), without requiring the presence of additional time–energy correlations; Supplementary Fig. 9 provides a more in-depth analysis of these. In this context, it is worth recalling that the propagation of a peeling front along an heterogeneous interface has been reported to be governed by irregular depinning jumps with power-law distributed sizes and inter-event times[24,26]. It might be interesting to check whether these jumps also form MS-AS sequences according to the fundamental seismic laws, and whether these actually relate to the individual distributions of energies and waiting times as anticipated here (Supplementary Eq. 8 for productivity law, Supplementary Eq. 12 and Supplementary Fig. 4 for Båth's law, Supplementary Eq. 18 for the Omori–Utsu law). Finally, our finding severely constrains, in this present situation of a single propagating crack, the design of probabilistic forecasting models for the occurrence, and energy of future AE events based on the scaling laws of seismology.

The origins of these laws can be discussed. Over the past years, the avalanche dynamics or crackling noise[37] exhibited by a tensile crack propagating in an heterogeneous solid has found a formulation in terms of a critical depinning transition of a long-range elastic manifold in a random potential[25,38–40]. Within this approach, the area swept by the crack front during a depinning event exhibits a scale-free distribution[25,41], along with the total elastic energy released within the sample during the event; in the nominally brittle fracture experienced here, these two quantities are proportional and the proportionality constant is equal to the fracture energy[27]. As a consequence, both productivity law and Båth's law are anticipated. Note, however, that there is no one-to-one relationship between the depinning events defined above and the acoustic events analyzed here. In particular, earlier work[27] has permitted, on the same artificial rocks, to measure the exponent $\beta'$ characterizing the scale-free distribution of depinning events. It was found to be significantly higher: $\beta' \simeq 1.4$ for $\bar{v} = 2.7$ μm s$^{-1}$. It is finally worth to mention that depinning models predict that, at vanishing driving rate, depinning events are randomly triggered in time, with exponential distribution for the inter-event time[42], in apparent contradiction with the scale-free distribution observed here on AE waiting times. However, it has been recently shown[42] how the application of a finite thresholding divides each true depinning avalanche into a correlated burst of disconnected sub-avalanches; the waiting times separating these correspond to the "hidden" parts below the threshold. A similar mechanism might be invoked here, where each depinning avalanche leads to a correlated burst of AE emitted by the successive points of strong acceleration/deceleration encountered during the avalanche considered.

The uncovering of the relations between the scale-free statistics of inter-event time and energy and the seismic laws characterizing the organization of AS sequences has been made possible since, in the experiments here, the time series are stationary. Surprisingly, the so-obtained relations are also compatible with observations reported on compressive fracture experiments: Our findings yield a relation $\alpha = \beta - 1$ between productivity and the Gutenberg–Richter exponent, which compares very well with what was observed in ethanol-dampened charcoal[21]: {$\beta = 1.30, \alpha = 0.28 \pm 0.01$}, in slowly compressed wood[20]: {$\beta = 1.40 \pm 0.03, \alpha \approx 0.3$}[20] and slowly compressed Vycor[19]: {$\beta = 1.40 \pm 0.05, \alpha = 0.33 \pm 0.07$}; Our findings also yield a prediction on how the magnitude difference, $\Delta M$, between the largest AS and its triggering MS depends on $\beta$ (Supplementary Fig. 4), and in particular that $\Delta M$ ($\beta = 1.3$) = 1, which is compatible with what was reported in ethanol-dampened charcoal[21]. In other words, the inter-relations between the seismic laws unraveled here in a single crack propagation situation and for a stationary time series seem to remain valid in the much more complex situations of compressional fracture, involving the collective nucleation of numerous microcracks and non-stationary time series. Conversely, the relation $\alpha = \beta - 1$ is not valid for earthquakes: For instance, analysis of the seismicity catalog for Southern California has yielded[12] $\beta = 1.72 \pm 0.07$ and $\alpha \approx 0.5$, which does not fulfill the relation $\alpha = \beta - 1$. Let us recall in this context that, in the Earth, the boundary

loading conditions are not forced as in lab experiments but may themselves be emergent properties from the fracturing system.

## Methods

**Synthesis of artificial rocks.** The artificial rocks were obtained by sintering polystyrene beads by means of the procedure described in[43] and briefly summarized herein. First, a mold filled with monodisperse polystyrene beads (Dynoseeds from Microbeads SA) of diameter $d$ was heated up to $T = 105 \,°C$ (90% of the temperature at glass transition). Second, a slowly linearly increasing compressive stress was applied while keeping $T = 105 \,°C$, up to a prescribed value $P$. Both $P$ and $T$ were then kept constant for one hour to achieve the sintering. Third, the system was unloaded and the sample was taken out of the mold, while keeping $T = 105 \,°C$ to avoid thermal shocks. Fourth, the sample was cooled down to ambient temperature at a rate slow enough to avoid residual stress. This procedure provides artificial rocks with homogeneous microstructure, the porosity and length-scale of which are set by $P$ and $d$, respectively. In all the experiments reported here, $P$ was chosen large enough (larger than 1 MPa) to have negligible rock porosity (less than 1%), regardless of $d$. This ensures a nominally brittle fracture with a single intergranular crack propagating in between the sintered grains (Supplementary Fig. 10). The heterogeneity length-scale is directly set by $d$. The heterogeneity contrast is mainly set by the small out-of-plane distortions due to the disordered nature of the grain joint network, which induces mixed mode fracture at the very local scale. The contrast hence remains small (weak heterogeneity limit), not sufficient to promote microcracking and quasi-brittle fracture[27,43]. For all experiments, except those underpinning Supplementary Fig. 8, the nominal diameter of beads prior sintering is $d = 583 \,\mu m$ and the standard deviation around is 28 μm. This diameter is large enough to ensure global crackling dynamics at finite driving rate[44]. In Supplementary Fig. 8, the first series of experiments was carried out with beads of nominal diameter $d = 233 \,\mu m$ and standard deviation 6.2 μm, and the second series of experiments with beads of nominal diameter $d = 24 \,\mu m$ and standard deviation 4 μm. Table 1 provides a synthesis of the samples and parameters to be associated with the different experiments analyzed here.

**Experimental arrangement for the fracture tests.** Stable tensile cracks were driven by the wedge splitting fracture set-up described in[27]. Parallelepiped samples of size $140 \times 125 \times 15 \, mm^3$ in the propagation, loading, and thickness directions were machined from the obtained artificial rocks. An additional $30 \times 40 \, mm^2$ rectangular notch was cut out on one of the two lateral edges and a 10 mm-long seed crack was introduced in its middle. This crack is loaded in tension by pushing a triangular steel wedge (semi-angle 15°) in the notch at a constant velocity, $V_{wedge}$. Two steel blocks with rollers coming in between the wedge and notch ensure the damage processes at the crack tip to be the sole dissipation source for mechanical energy in the system. During each experiment, the force $f(t)$ applied by the wedge was monitored in real-time by means of a S-type Vishay cell force, and the instantaneous specimen stiffness $k(t) = f(t)/V_{wedge} \times t$ was deduced. From this signal and the knowledge of the variation of $k$ with crack length $c$ in such a geometry (obtained by finite element simulations), the instantaneous crack length (spatially averaged over specimen thickness) was obtained and the instantaneous spatially averaged crack speed was deduced (see[27] for details). Its mean value, $\bar{v}$ over the considered range for acoustic analysis was tuned by modulating the wedge speed.

**Monitoring of acoustic events.** The acoustic emission was collected at 8 different locations via 8 piezoacoustic transducers. The signals were preamplified, band filtered, and recorded by a PCI-2 acquisition system (Europhysical Acoustics) at 40 MSampless$^{-1}$. An acoustic event (AE) is defined to start at the time $t_i$ when the preamplified signal $V(t)$ goes above a prescribed threshold (40 dB), and to stop when $V(t)$ decreases below this threshold. The minimal time interval between two successive events is 402 μs. This interval breaks down into two parts: The hit definition time (HDT) of 400 μs and the the hit lockout time (HLT) of 2μs. The former sets the minimal interval during which the signal should not exceed the threshold after the event initiation to end it and the latter is the interval during which the system remains deaf after the HDT to avoid multiple detections of the same event due to reflexions. Each so-detected AE is characterized by two quantities: Its occurrence time identified with $t_i$ and its energy defined as the square of the maximum value $V^2(t)$ between $t_i$ and $t_f$; we have verified that the results reported here do not change if we choose instead to define the energy as the integral of $V^2(t)$ over the duration of the event (Supplementary Fig. 11). From the knowledge of the wave speed, $c_W$, in the material (measured using the pencil lead break procedure: $c_W = 2048 \, ms^{-1}$) and the arrival time at each of the 8 transducers, it is also possible to localize spatially the sources of emitted AE (Supplementary Movie 1). The spatial accuracy is set by the main frequency $f$ of the AE waveform. This frequency was measured to vary from $f = 40 \,kHz$ to $f = 130 \,kHz$ depending on the analyzed pulse, yielding an overall spatial accuracy $\delta x \sim f/c_W \sim 5 \, mm$.

**Data availability.** The data that support the findings of this study are available from the corresponding author upon reasonable request.

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

## Acknowledgements

We thank Thierry Bernard for technical support, and Hervé Bercegol and Vadim Nikolayev for fruitful discussions. We also thank Francois Daviaud, Hugues Chaté, Francois Ladieu, Cindy Rountree, and Julien Scheibert for the careful reading of the manuscript. Funding through ANR project MEPHYSTAR (ANR-09-SYSC-006-01) and by "Investissements d'Avenir" LabEx PALM (ANR-10-LABX-0039-PALM) is also gratefully acknowledged.

## Author contribution

J.B., D.D., and D.B. conceived the experiment; J.B., M.L.H., D.D., and D.B. performed the experiments; J.B. A.D., and D.B. analyzed the data; D.B. wrote the manuscript; all authors reviewed the manuscript.

## Additional information

**Competing interests:** The authors declare no competing interests.

