## [Peer Review File · Nature Communications]

Reviewers' comments:

Reviewer #1 (Remarks to the Author):

Aftershock sequences and seismic-like organization of acoustic events produced by a single propagating crack

By Jonathan Barés, Alizée Dubois, Lamine Hattali, Davy Dalmas & Daniel Bonamy.

This paper is an important contribution to the understanding of the origin and nature of power-law scaling during quasi-brittle failure in heterogeneous media. One of the most important results is that aftershocks observed are not due to spatiotemporal correlations, but instead to the emergent properties of the underlying scaling rules under the stationary loading conditions examined here. It is generally very well written and succinctly presented, sticking to the most important points and providing a clearly logical narrative.

I recommend publication, subject to addressing a few comments, predominantly on context. Some additional more minor ones are indicated on the annotated manuscript.

1. I think it is important in the abstract and introduction to present the results as investigation into an alternate hypothesis to the conventional 'emergent' behaviour from a more complex medium. In fact, the results presented here are essentially one end-member of what could happen in quite a large phase space of variables (tensile v. compressional, slow v. fast loading, smaller v. large material disorder, 'starting notch' v. emergent localisation). I don't think the authors are arguing the results apply outside the chosen parameters, but some of the claims here make it look as if they are, as clarified later in the discussion and in the supplementary material, where the language is more accurate and the context clearer.

2. Introduction. In previous work on the effect of heterogeneity on brittle to quasi-brittle behaviour, it is quite clear the starting porosity is an important control [e.g. Vasseur, J. et al. (2015). *Sci. Rep.* 5, 13259], itself dependent on the variability of diameter about the mean using synthetic materials. For a sample of beads of one size, porosity is independent of diameter, so it is not clear how diameter controls 'heterogeneity' per se. What is the standard deviation of the beads in the experiments for example? Here the authors pick a diameter for pragmatic reasons – i.e. a diameter large enough to produce easily-detectable Acoustic Emissions? One might expect materials with larger diameter particles of equal size to produce more dynamic rupture events that would be more easily detected.

3. (First para of results section). It is true the loading geometry, including the notch has a big effect on concentrating the stress and encouraging a single crack to propagate is true. However, quasibrittle failure (even with notch specimens in tension under stationary loading), with cracking nevertheless distributed throughout the sample, can be promoted more by slower loading conditions and/or a more active chemical environment [see e.g. Hatton et al., (1993) *J. Struct. Geol.* 15, 1485-1495].

Editorial Note: Annotated Manuscript by Reviewer 1 unable to be reproduced as part of this Peer Review File.

Reviewer #2 (Remarks to the Author):

This work looks at the Acoustic Emission (AE) events in the propagation of a single crack in a heterogeneous material. The AE events form a timeseries, of which observations are made. These observations match a number of observations from earthquakes ("laboratory earthquakes").

I have three problems with this work, that in my opinion are contrary to publishing it in Nature Communications.

First of all, I am missing the idea of the "self-organization" the authors make a point about. I think that they rather argue (energies of AE events in particular or the waiting time distribution of events) that there is not much organization at all. I think this is a misleading way of presenting the results.

The second is that the Oslo group planar crack experiments have demonstrated a large number of these results in a similar setting (in-plane single crack vs. this case where out-of-plane crack excursions also take place). This puts, without the in-depth commentary the manuscript is missing, the novelty of some of the conclusions at doubt.

The third is that this kind of dynamical crack propagation problems are supposed to derive their scaling from the depinning of cracks as elastic manifolds in a random medium - as the Oslo case has shown convincingly in all aspects. Here, while the scale-free distribution of the AE energies follows a power-law as expected, the waiting times do so as well, as is not expected. The authors do note this, in the discussion, and provide three candidate reasons for this discrepancy. I think that this work can not be published without actually explaining this or trying to do so. In particular there are ways of even testing these three ideas (ref. 26 for instance discusses similar issues already!).

One would also think that the authors could probe more deeply into the possible spatiotemporal correlations in the AE timeseries (some such should exist, since the waiting time distribution is not Poissonian).

Other comments:

The authors argue, that the rate R is constant, possibly so, but the timeseries would be interesting anyways.

Is the beta-exponent of $P(E)$ known from earlier work? Or explained by crack depinning?

The physical meaning of Equation 3 would merit a comment.
Figure 2b misses unit (time).

What is the real time accuracy of the AE system? Milliseconds, microseconds?

Can one translate the integral of AE energy into crack length? I.e. why should the event number be a good quantity to integrate? Or, better than that integral.

Reviewer #3 (Remarks to the Author):

Comments on "Aftershock sequences and seismic-like organization of acoustic events produced by a

single propagating crack (NCOMMS-17-04297-T)” By J. Bares, A. Dubois, L. Hattali, D. Dalmás, and D. Bonamy

This article presents a statistical analysis of the observational data of acoustic emissions from loading some artificial rocks (made by sintering polystyrene beads) with a signal open crack. The results show the observations can explain: 1) The G-R law for earthquake energies (magnitudes), 2) Aftershock productivity law; 3) The Omori-Utsu formula; 4) the Bath law. This study is an excellent work, and potentially acceptable by Nature Communications. For my viewpoints, there are some important issues need to be tackled in the revision.

1. (Lines 3 to 14, Page 7; Lines -3, Page 4 to 10, Page 5, in supplementary material) First of all, the derivation of the productivity law of the aftershock, i.e, Equations (3) and (S7), is strange. NAS(EMS) defined here is the ratio of the number of events smaller than the mainshock to that of events bigger than the mainshock in the whole catalog.

From Figure 2a, we can see that β is about 1. If so, α is about 0, meaning that there is no difference in aftershock productivity among mainshocks of different magnitudes.

I suggest that you use some declustering methods (von Stiphout, 2019) to get earthquake clusters and then estimate NAS(EMS) based on the size of the clusters.

2. The Bath law. There have been many literature of the explanation of the Bath law to explain that the Bath law is a corollary of the two exponential laws: the negative exponential law for magnitudes and the position exponential law for aftershock productivity. Please see Luo and Zhuang (2016) and references therein.

3. (Lines 6 to 14, Page 10) In the detection of earthquakes, after a big earthquake, many small earthquakes are missed from the catalog. This is because the duration of the waveforms generated by the mainshock and some large aftershocks dominate the recorded seismograph, and thus the waveform of small earthquakes cannot be recognized. Do your sensors have the similar problems?

4. Equation (4). Is the tapered tail caused by the limitation of observation window?

Minors

1. Equation (1), please specify whether $P(E)$ is the probability density function or the cumulative probability distribution function. Also, what is the relation between your β and the G-R b-value?

2. Line 7, Page 9. How did you do “permuting randomly”?

3. Figure 1. Please give a plot of cumulative numbers of events against time.

4. Figure 3B. In most of results of the Bath law, the magnitude difference between the mainshock and the largest aftershock increases with the mainshock magnitude. But your plot seems gives an contrary result. Can you please give some explanation?

References:

van Stiphout T., Zhuang J., and Marsan D. (2012) Seismicity declustering, Community Online Resource for Statistical Seismicity Analysis, doi:10.5078/corssa-52382934. Available at <http://www.corssa.org>.

Luo, J. and J. Zhuang (2016) Three regimes of the distribution of the largest event in the critical ETAS model. Bulletin of the Seismological Society of America. 106(3), 1364-1369. doi:10.1785/0120150324.

We would like to thank reviewer 1 for his/her valuable comments and questions. We modified our manuscript to reply his various concerns. We listed below the detailed response (in black) to the different points raised by the reviewer (in blue italic). The changes yielded in the manuscript are specified accordingly. They appear in red in the manuscript and its supplementary materials.

1. I think it is important in the abstract and introduction to present the results as investigation into an alternate hypothesis to the conventional 'emergent' behaviour from a more complex medium. In fact, the results presented here are essentially one end-member of what could happen in quite a large phase space of variables (tensile v. compressional, slow v. fast loading, smaller v. large material disorder, 'starting notch' v. emergent localisation). I don't think the authors are arguing the results apply outside the chosen parameters, but some of the claims here make it look as if they are, as clarified later in the discussion and in the supplementary material, where the language is more accurate and the context clearer.

Our experiments indeed show that the organization in aftershock sequences for AE and the associated seismic laws (the productivity law, the Bath law and the Omori-Utsu law) are not specific to the multi-cracking situations of compressive quasi-brittle fracture or seismology, but are already present in the (much simpler) situation of a single slowly propagating tensile crack. As such, the additional complexity brought by the interactions between multiple microcracks is not a mandatory ingredient to observe them. Conversely, such additional element of complexity may significantly change the picture.

We change the abstract so as to make it clearer.

2. Introduction. In previous work on the effect of heterogeneity on brittle to quasi-brittle behaviour, it is quite clear the starting porosity is an important control [e.g. Vasseur, J. et al. (2015). Sci. Rep. 5, 13259], itself dependent on the variability of diameter about the mean using synthetic materials. For a sample of beads of one size, porosity is independent of diameter, so it is not clear how diameter controls 'heterogeneity' per se. What is the standard deviation of the beads in the experiments for example? Here the authors pick a diameter for pragmatic reasons – i.e. a diameter large enough to produce easily-detectable Acoustic Emissions? One might expect materials with larger diameter particles of equal size to produce more dynamic rupture events that would be more easily detected.

Here, we choose the sintering conditions to have a porosity as low as possible (less than 1%), irrespectively of the bead diameters. This condition is required to ensure a nominally brittle fracture driven by a single dominant opening crack. This is made possible by applying a compressive stress, P , large enough during the hot phase of the sintering process ($P \geq 1\text{MPa}$). This point is precised in the revised version ("Method" section, pp. 16). In particular, a photo showing the fracture surfaces, its facet-like structure illustrating the intergranular fracture mode and the absence of visible porosity has been added (supplementary Fig. S8).

Heterogeneities in our sintered materials are characterized by two parameters: Their length scale and their contrast (mainly given by the facet orientations). The length-scale is directly set by the diameter of the beads used in the sintering process and, since we used monodisperse beads (see below), we had an accurate control on this length-scale. Conversely we do not have any control on the heterogeneity contrast. Modulating the porosity would have permitted to modulate this one but we chose to keep the porosity close to zero to ensure a nominally brittle fracture mode. As the crack was shown to be intergranular and to propagate in between the sintered particles (see new supplementary Fig. S8 & ref 49), this contrast is mainly set by small out-of-plane distortions of the front with respect to a planar plane, due to the disordered nature of the grain joint network, which induce mixed mode fracture at the

very local scale. The contrast hence remains small and we stay in the limit of weak heterogeneities. This point is now specified in the revised manuscript (“Method” section, pp. 16).

In our experiments, the artificial rocks were built from monodisperse beads: For most of the experiments (all figures except Fig. S7 in supplementary materials), the nominal diameter is $d=583\mu\text{m}$ and the standard deviation was $14.41\mu\text{m}$ (provided by the manufacturer). In Fig. S7 -- supplementary materials, the first series of experiments were carried out with beads of nominal diameter $d=233\mu\text{m}$ (not $250\mu\text{m}$ as initially thought) and standard deviation of $6.2\mu\text{m}$, and the second series of experiments with beads with nominal diameter $d=24\mu\text{m}$ (not $20\mu\text{m}$ as initially thought) and standard deviation $4\mu\text{m}$. This information is now provided in the revised manuscript (“Method” section, pp. 16).

In the main text of the manuscript, we choose indeed to analyze the statistics of AEs keeping the same bead size (the largest one: $d=583\mu\text{m}$). Conversely, we checked that changing (reducing) the bead size do not change the picture : The organization into aftershock sequences obeying Gutenberg-Richter, Omori, productivity and Bath laws remains true when $d=233\mu\text{m}$ and $d=21\mu\text{m}$. This is mentioned in the manuscript. It is worth to see that changing d changes the value of the associated exponents (see Fig. S7 in supplementary materials). In particular, the RG exponent is $\beta \sim 1 - 1.1$ for $d=233\mu\text{m}$ and $\beta \sim 1.3 - 1.4$ for $d=21\mu\text{m}$, and the waiting time exponent is $\gamma \sim 1.4 - 1.9$ for $d=233\mu\text{m}$, and $\gamma \sim 1.1 - 1.6$ for $d=21\mu\text{m}$ (in both case depending on the crack speed). This permits to demonstrate that these exponents are not only velocity dependent, but also material dependent. This point is now specified in the revised manuscript (pp. 12).

3. (First para of results section). It is true the loading geometry, including the notch has a big effect on concentrating the stress and encouraging a single crack to propagate is true. However, quasi- brittle failure (even with notch specimens in tension under stationary loading), with cracking nevertheless distributed throughout the sample, can be promoted more by slower loading conditions and/or a more active chemical environment [see e.g. Hatton et al., (1993) J. Struct. Geol. 15, 1485-1495].

We agree with that. Here, the experimental conditions were achieved to ensure nominally brittle fracture driven by the propagation of a single dominant crack, even at slow speeds (see response to point 2). The fact that quasi-brittle is promoted against nominally brittle by slower loading rate and more activated environment is now mentioned in the introduction (pp. 2).

The minor points indicated in the manuscript are addressed below :

4/ Abstract : At a more fundamental level this in turn will be controlled by the degree of heterogeneity in the material [Vasseur, J. et al. (2015). Sci. Rep. 5, 13259] strain rate [Ojala I O. et al. (2004), Geophys. Res. Lett., 31, L24617, etc. Maybe worth mentioning this in the Introduction?

This is done in the revised version of the manuscript (see introduction, pp. 2)

5/ Abstract : I wonder about the word 'single' here. Maybe 'single dominant' (as above) or 'dominant' is better, since there will normally be other much smaller micro-cracks, predominantly in the process zone ahead of the dominant crack tip

This is true. The terminology « produced by a single nominally brittle propagating crack » has been replaced by « produced by a single dominant crack ». At the end of the abstract also, the terminology « A single slowly moving opening crack » has been replaced by « The nominally brittle one, driven by the propagation of a single dominant opening crack »

6/ Introduction : 'conjectured' is a strong word to use here. It might be fairer to say the observed behaviour is found to be consistent with emergent properties of a disordered system in some physical models of heterogeneous media, but this does not preclude other interpretations.

True. The sentence has been replaced by : « They are usually seen as emergent properties for the collective dynamics of microcrack nucleation, structured by the long-range stress redistribution following each microfracturing event. » (introduction, pp. 3).

7/ Introduction : This is a good alternate hypothesis to test. However, it is more likely an end-member for materials which will tend to be more homogeneous, operating at high strain rate or where there is a strong pre-existing stress concentrator, with the actual behaviour often at some intermediate level.

This is the scenario we test : Can we find similar « seismic-like » time-energy organization for acoustic events in the simpler (and more tractable) problem of nominally brittle fracture in heterogeneous solids, driven by the propagation of a single dominant crack.

The conditions to achieve such a nominally brittle fracture is indeed to have weak heterogeneities (and in particular no porosity) and a pre-existing strong stress concentrator (a precrack). Our experiment was designed to achieve these two conditions. Note however that, when these two conditions are fulfilled and nominally brittle fracture mode is selected, a very slow crack speed is required to observe crackling (i.e. highly intermittent crack growth made of sudden random impulses of broadly distributed sizes, see ref. [50]).

The manuscript has been changed to make it clearer (see response to point 1 and 2 raised by the reviewer and induced changes in the manuscript and abstract pp. 1 & “Method” section, pp 16).

8/ Introduction: This is interesting - changing the size of the beads does not change the porosity per se, unless the beads themselves have a range of sizes in each case. In other studies the porosity is the main control on more brittle or more semi-brittle behaviour [e.g. Vasseur, J. et al. (2015). Sci. Rep. 5, 13259]

Here, we choose the sintering conditions to have a porosity as low as possible (less than 2%), irrespectively of the bead diameters. This condition is required to ensure a nominally brittle fracture driven by a single dominant opening crack (see response to point 2 and induced modification in the manuscript, “Method” section pp. 16).

This is made possible by applying a compressive stress, P , large enough during the hot phase of the sintering process ($P > 1\text{MPa}$). This point is specified in the « Methods » section (pp 16) of the revised version (see response to point 2).

9/ Results : 'The productivity'

This has been corrected (“Results”, pp. 7)

10/ Results : Yes, as do the scaling exponents in Hatton et al., 1993.

This is now mentioned (“Results” pp. 11)

11/ Discussion : Maybe you could discuss that this was forced by the experimental design. In the Earth the boundary conditions would be quite different, and may themselves be emergent.

This is done in the revised version (“Discussion” pp.15)

We would like to thank reviewer 2 for the valuable comments and questions. We listed below the detailed response (in black) to the different points raised by the reviewer (in blue italic). The changes yielded in the manuscript are specified accordingly. They appear in red in the manuscript and its supplementary materials.

1/ First of all, I am missing the idea of the "self-organization" the authors make a point about. I think that they rather argue (energies of AE events in particular or the waiting time distribution of events) that there is not much organization at all. I think this is a misleading way of presenting the results.

We disagree with the reviewer on this point. There is an organization of the AEs into aftershock sequences obeying scaling laws reminiscent of those observed in seismology : Productivity law & Eq. 3, Bath law, Omori laws & Eq. 4. Neither of these laws will be fulfilled for non-organized events, i.e. a Poisson process of successive events with « standard » (i.e. not-scale free) energy distribution. This specific organization is demonstrated in our manuscript to be a consequence of the scale-free distribution of both AE energy and interevent times.

For this reason, it is not misleading to speak about the organization of the AEs (even if in our case, the organization does not come from correlations between time occurrence and energy or correlation between the energy of successive events but comes from the scale free distribution of energy and waiting time as we demonstrated). In the initial version of the manuscript, we call it a « self-organization ». In the revised version, we drop the « self-« which can be misleading (possible confusion between self-organization and self-criticality) : « self-organize » has been replaced by « get organized » and « self-organization » has been replaced by « organization » (abstract pp. 1 & 2).

2/ The second is that the Oslo group planar crack experiments have demonstrated a large number of these results in a similar setting (in-plane single crack vs. this case where out-of-plane crack excursions also take place). This puts, without the in-depth commentary the manuscript is missing, the novelty of some of the conclusions at doubt.

We also disagree with the reviewer on this point. It is worth to recall here the two main outcomes of our work:

1. The fact that, in a so situation of nominally brittle fracture driven by the propagation of a single crack, AE organize into mainshock-aftershock events obeying the most common seismic laws : Omori-Utsu law (aftershock frequency decays algebraically with time from mainshock), productivity law (number of produced aftershocks increases as a power-law with mainshock energy), and Bath law (difference in magnitude between mainshock and its largest aftershock is independent of the mainshock magnitude).
2. The fact that the above seismic laws for the AS organization directly result from the scale-free statistics of energy (for productivity law and Bath law) and from that of inter-event time (for Omori law), and are not due to time-energy correlations (or spatiotemporal correlations). This is demonstrated in the present manuscript and permit to predict some interrelations between the parameters at play in the seismic laws and that of the scale-free statistics.

To the very best of our knowledge, neither of these results have been reported on the Oslo group planar crack experiments.

It is also worth mentioning that the two setting (ours and Oslo one) are not similar. At least three differences of significant importance have to be mentioned : The dimensionality (2D planar crack in their case vs 3D fracture in ours), the nature of the loading (mixed mode in Oslo case, du to the

asymetry of the loading vs pure mode I in our case) and the studied observables (depinning avalanches in Oslo case vs. acoustic events in ours, see also response to points 3 & 6).

3/ The third is that this kind of dynamical crack propagation problems are supposed to derive their scaling from the depinning of cracks as elastic manifolds in a random medium - as the Osloc case has shown convincingly in all aspects. Here, while the scale-free distribution of the AE energies follows a power-law as expected, the waiting times do so as well, as is not expected. The authors do note this, in the discussion, and provide three candidate reasons for this discrepancy. I think that this work can not be published without actually explaining this or trying to do so. In particular there are ways of even testing these three ideas (ref. 26 for instance discusses similar issues already!).

It is important to emphasize the differences between AE and depinning avalanches (see also response to point 6). They are not the same objects:

- Depinning avalanches are elastostatic quantities. They correspond to the events when the crack line depins from a heterogeneity and, by doing so, releases part of the elastic (elastostatic) energy stored in the sample by creating a given amount of fracture surface (in the standard depinning framework, the avalanche size is actually defined by this depinning area);
- AE are elastodynamics quantities : they are the signature of the elastic waves triggered by the local accelerations/decelerations going along with the above depinning events ? These elastic waves are (locally) collected by the transducers placed on the sample.

There is no one-to-one relation between the two objects. In particular, the AE energy is not proportional to the size of the depinning events. In this context, the discussion section of the first verison of the manuscript was misleading. It has been significantly rewritten to emphasize this point (pp. 13).

The fact that the paradigm of depinning elastic manifolds would rather yield to Poissonian waiting time is briefly noted in the discussion of the manuscript, the candidate reasons in its adaptation to the fracture problem that may turn the waiting time distribution to a power-law also. Still, this is clearly not the focus of this manuscript (see response to point 2 which recalls the two main outcomes of our work).

4/ One would also think that the authors could probe more deeply into the possible spatiotemporal correlations in the AE timeseries (some such should exist, since the waiting time distribution is not Poissonian).

The spatial localization accuracy of the AE sources is not sufficient to permit such an analysis. This spatial accuracy δx is set by the main frequency of the AE waveform, $40kHz \leq f \leq 130kHz$ and the wave speed in the material, $c_w = 2048m/s$ (for the artificial rock made of 583 μm PS beads used in the experiments having lead to Figs 2 and 3) : $\delta x \sim \frac{f}{c_w} \sim 5 mm$ to be compared with the specimen thickness of 1.5cm.

The spatial localization accuracy and the way it can be estimated are now provided in the revised manuscript (pp. 17 & 18, section « Methods »).

5/ The authors argue, that the rate R is constant, possibly so, but the timeseries would be interesting anyways

We do not say that $R(t)$ [the number of AE produced per unit time] is constant. In our experiment, we demonstrate that it is the number of AE produced per unit length which is constant (see also response to point 10). As a result, $R(t)$ is proportional to the instantaneous velocity $v(t)$ (with a proportionality constant $\sim H/d^2$ where H is the specimen thickness and d the microstructure length scale, see text), and the mean value \bar{R} is proportional to the mean value \bar{v} over the considered time window.

In the revised manuscript, a figure showing the cumulative number of AE as a function of time has been added (Inset in Fig. 1D). A figure showing the variation of \bar{R} with \bar{v} has also been added in the supplementary materials (Fig. S4).

6/ Is the beta-exponent of $P(E)$ known from earlier work? Or explained by crack depinning?

This is not the case. Here, E is the energy of an acoustic event as it arrives at the piezoelectric transducer, defined as the square of the maximum value $V^2(t)$ of the preamplified voltage $V(t)$ measured on the piezo-acoustic transducer over the duration of the event [It was checked that defining the energy as the integral of $V^2(t)$ over the duration of the event does not change the value of beta]. This acoustic energy is very different from the total elastic energy released by a depinning event, *i.e.* when the crack line locally depins and progresses over one unit: It is orders of magnitude smaller. More importantly, this acoustic energy has no reason to be proportional to the total elastic energy released by a depinning event : the waveform associated with the acoustic pulse will depend on the depinning event, but also on the complete geometry of the specimen at the time of the event, the eigenmodes at that times, their spatial distribution and how they are perceived at the location of the transducer...

Actually, we did measure in an earlier work the distribution of the energy released by the depinning events in the same experiments; the results of the study were reported in Ref. 27. In this case, the energy release was shown to be proportional to the area swept by the depinning avalanches and, as such, has been compared to the predictions of the depinning theory of elastic manifolds. The exponent β' to be associated with this true energy is found:

- To be significantly larger than that measured for the acoustic energy : $\beta' \sim 1.4$ for $v = 2.7 \mu\text{m/s}$, see Ref. 27 Fig. 3A (empty symbols associated with $V_{\text{wedge}} = 16 \text{nm/s}$) to be compared with $\beta \sim 0.96$ for the acoustic energy (fig. 2A of the present manuscript);
- To significantly depend on the mean crack speed, much more than β : $\beta' \sim 1.4$ for $v = 2.7 \mu\text{m/s}$ ($V_{\text{wedge}} = 16 \text{nm/s}$ empty symbols in Fig. 3A of Ref. 27) and $\beta' \sim 1.1$ for $v = 27 \mu\text{m/s}$ ($V_{\text{wedge}} = 16 \text{nm/s}$) filled symbols in fig. 3A Ref. 27) Fig. 3A, filled symbols. As $\beta \sim 0.96$ in the first case, and $\beta \sim 0.93$ in the second case.

The fact that the acoustic energy measured here is very different from the energy released by a depinning event is now emphasized in the manuscript (« Discussion » pp. 13). The fact that the exponent β for the acoustic energy is different from that of true (elastostatic) energy released by the depinning events is also specified (« Discussion pp. 13).

7/ The physical meaning of Equation 3 would merit a comment.

The derivation of Equation 3 is based on the fact that the event energies are independent with respect to each others and not correlated with the time occurrence. The derivation is detailed page 5 in supplementary materials. The total number of events with an energy smaller that the prescribed energy E_{MS} for mainshock gives the total number of aftershock (AS) in the catalog (summed over all AS

sequences). The total number of event with an energy larger than E_{MS} gives, by definition, the total number of mainshock (MS), and hence the total number of AS sequences. The ration between the two, hence, gives the mean number of AS per sequence which is, by definition, $N_{AS}(E_{MS})$.

The associated text pp. 5 in supplementary materials has been reformulated to make it clearer.

8/ *Figure 2b misses unit (time).*

This has been corrected.

9/ *What is the real time accuracy of the AE system? Milliseconds, microseconds?*

The acquisition rate is 40MSample/s. It should also be noted that the minimal interval between two successive events is $402\mu\text{s}$. This duration breaks down into two parts :

- Hit Definition time (HDT) of $400\mu\text{s}$, which is the minimal interval during which the signal should not exceed the threshold after an event initiation to end it ;
- Hit Lockout time (HLT) of $2\mu\text{s}$, which is the interval during which the system remains deaf after the HDT to avoid multiple detections of the same event due to reflexions. In our case, the latter has been reduced to the minimum value available in our system (i.e. $2\mu\text{s}$) due to the small size of ours sample.

These information are now provided in the revised manuscript (« methods » section, pp. 17 & 18).

10/ *Can one translate the integral of AE energy into crack length? Ie. why should the event number be a good quantity to integrate? Or, better than that integral.*

The integral of AE energy cannot be translated into crack length because of the arguments developed in response to point 6. In our experiments, we do observe that the cumulative number of events increase linearly with the crack length, irrespectively of the crack speed. This is an experimental observation. Such observation can be interpreted by stating that the number of AE produced as the crack propagates over a unit length is given by the number of heterogeneities met over this period. Then, the cumulative number of heterogeneities linearly grows with crack length, and so does the cumulative number of produced AE (see Fig. 1D).

If now we plot the integral of the AE energy as a function of crack length (instead of the integral of the number of AEs), we do not observe a straight line (see figure below). This shows that the AE energy is not the good quantity to integrate.

Cumulative AE energy as a function of crack length for $\{d = 583\mu\text{m}, \bar{v} = 2.7\mu\text{m/s}\}$

We would like to thank reviewer 3 for his/her very valuable comments and questions. We modified our manuscript to reply his/her various concerns. We listed below the detailed response (in black) to the different points raised by the reviewer (in blue italic). The changes yielded in the manuscript are specified accordingly. They appear in red in the manuscript and its supplementary materials.

1/ (Lines 3 to 14, Page 7; Lines -3, Page 4 to 10, Page 5, in supplementary material) First of all, the derivation of the productivity law of the aftershock, i.e, Equations (3) and (S7), is strange. $N_{AS}(E_{MS})$ defined here is the ratio of the number of events smaller than the mainshock to that of events bigger than the mainshock in the whole catalog.

The derivation of Equation 3 is based on the fact that the event energies are independent with respect to each others and not correlated with the time occurrence. The derivation is detailed page 5 in supplementary materials. The total number of events with an energy smaller than the prescribed energy E_{MS} for mainshock gives the total number of aftershock (AS) in the catalog (summed over all AS sequences). The total number of event with an energy larger than E_{MS} gives, by definition, the total number of mainshock (MS), and hence the total number of AS sequences. The ratio between the two, hence, gives the mean number of AS per sequence which is, by definition, $N_{AS}(E_{MS})$.

The associated text pp. 5 in supplementary materials has been reformulated to make it clearer.

2/ From Figure 2a, we can see that β is about 1. If so, α is about 0, meaning that there is no difference in aftershock productivity among mainshocks of different magnitudes.

Equation S8 (in supplementary materials) which predicts $\alpha = \beta - 1$ is valid when β is strictly superior to 1 and in absence of exponential cutoff. In figure 2A, neither of these two assumptions are true. Hence, we should apply Equation S7, valid for any distribution, even if $\beta \approx 1$ (or smaller) and in the presence of an exponential cutoff. And this formula works extremely well : The prediction of Equation S7, provided by the red line in Figure 2A, coincides almost exactly with the experimental points.

3/ I suggest that you use some declustering methods (von Stiphout, 2019) to get earthquake clusters and then estimate $N_{AS}(E_{MS})$ based on the size of the clusters.

We thanks the reviewer for pointing this reference out. It should be noted that most of the declustering methods proposed there require the spacial localization of the event source. This spatial information is not available in our experiments: This spatial accuracy δx is set by the main frequency of the AE waveform, $40kHz \leq f \leq 130kHz$ and the wave speed in the material, $c_W = 2048m/s$ (for the artificial rock made of $583\mu m$ PS beads used in the experiments having lead to Figs 2 and 3) : $\delta x \sim f/c_W \sim 5mm$ to be compared with the specimen thickness of 1.5cm. As a result, Gardner & Nnopoff windowing technique, SLC & Reasenberg declustering algorithms & Zhang et al & MISD stochastic declustering algorithms, which are all based on the spatio-temporal proximity of the events to the others, cannot be applied in our case. This limitation is now explicitly indicated in the revised version of the manuscript (« Results » pp. 7).

Van Stiphout et al. (2012) also report three declustering methods based on the analysis of the occurrence time only, which could *a priori* be applied to our experiments : Frohilch & Davis (1985) method, where each event is labelled MS or AS depending on the the ratio T_{Na}/T_{Nb} between the occurrence time T_{Na} of the N_a^{th} subsequent event and that T_{Nb} of the N_b^{th} previous event ; Bottiglieri et al (2009) method, which makes use of the variability coefficient (ratio between standard deviation and average value of Δt) to identify the AS sequences ; and a method derived from the work of

Haintz et al. (2006) which compute from the value of inter-event time the probability that the two consecutive events are from two different sequences or not. These three methods are very similar; the idea is to identify the AS sequences with groups of events in which the occurrence times depart from what is expected for a Poisson process. We applied the last method to our data and we estimated $N_{AS}(E_{MS})$ and $N_{FS}(E_{MS})$ based on the size of the AS clusters (supplementary Fig. S5). The so-obtained curves are pretty similar to that we obtained before. Note that, as mentioned by van Stiphout et al. (2012), such a declustering method do not exploit important information when searching for the AS sequence, namely the event energy.

4/ The Bath law. There have been many literature of the explanation of the Bath law to explain that the Bath law is a corollary of the two exponential laws: the negative exponential law for magnitudes and the position exponential law for aftershock productivity. Please see Luo and Zhuang (2016) and references therein.

We thank the reviewer for pointing out this part of the literature. In epidemic-type aftershock sequence (ETAS) models, the catalogs are built using a stochastic branching process in which an event (mother) gives birth to a number of offspring events (daughters) at a prescribed rate chosen to respect GR, productivity and Omori laws. In such approach, the Bath law emerges from the specific sequencing of the events (e.g. the energy correlation between successive events) caused by the branching process.

The situation is qualitatively different in our system where there is no correlation 1) between the energy of successive events and 2) between the energy and time occurrence. In this latter case, the Bath law can be derived using simple arguments of extreme event statistics, as proposed in supplementary materials.

This difference between our situation and ETAS model is now pointed out in the revised version of the manuscript (« Results » pp. 8 & 9).

5/ (Lines 6 to 14, Page 10) In the detection of earthquakes, after a big earthquake, many small earthquakes are missed from the catalog. This is because the duration of the waveforms generated by the mainshock and some large aftershocks dominate the recorded seismograph, and thus the waveform of small earthquakes cannot be recognized. Do your sensors have the similar problems?

A similar mechanism is likely at play in our system. Indeed, the characteristic time setting the dead-time after the mainshock is found to be very close to the duration of the largest acoustic events, irrespectively of the crack speed. This strongly suggest that the small ASs immediately following a (largest) mainshock are lost since their waveform are drown in that of the mainshock. This mechanism is indeed analog to the problem of short time aftershock incompleteness (STAI) encountered in seismology and known to be responsible of important biases in the estimation of the so-called c-value of Omori law (this c-value in seismology is to be mapped with τ_{min} in the present study).

This point is added to the manuscript (« Discussion » pp. 11 & 12).

6/ Equation (4). Is the tapered tail caused by the limitation of observation window

True, see response above.

7/ Equation (1), please specify whether $P(E)$ is the probability density function or the cumulative probability distribution function. Also, what is the relation between your beta and the G-R b-value?

In equation (1), $P(E)$ is the probability density function for energy. This is now specified in the revised version (« Results », pp. 5).

Gutenberg-Richter (G-R) frequency-magnitude relation considers the distribution of magnitude, M , rather than that of energy, E . M is linearly related to the logarithm of E : $\log_{10}E = 1.5M + 11.8$ for earthquakes, see Kanomori J. Geophy. Res. 82, 2981—2987 1977. The power-law probability density function for energy then leads to the classical G-R frequency magnitude relation $N(M) \propto a - bM$ where the G-R b value relates to β via : $\beta = b/1.5 + 1$.

This is now specified in the revised version (« Results » pp. 5, footnote #1)

8/ Line 7, Page 9. How did you do “permuting randomly”?

Each acoustic event (AE) is characterized by two quantities : its time occurrence and its energy. By permuting the AE energy, we means that randomly associated the energy of a given event with the time occurrence of another event.

This is now specified in the revised version of the manuscript (« Results » pp. 7).

9/ Figure 1. Please give a plot of cumulative numbers of events against time.

The curve is now provided in inset of Figure 1D.

10/ Figure 3B. In most of results of the Bath law, the magnitude difference between the mainshock and the largest aftershock increases with the mainshock magnitude. But your plot seems gives an contrary result. Can you please give some explanation?

In our case, the Bath law directly results from the probability density function of energy, $P(E)$ (see Eq. S10 in Supp. Mat.). As a result, the form of the curve depends on the precise form of $P(E)$. When $P(E)$ is a power-law with an exponent β strickly superior to 1 and in absence of exponential cutoff (standard G-R law), the Bath law takes a simpler form (see Eq. S11 in Supp. Mat.) which predicts a magnitude difference which slightly increases with the MS amplitude and converge to a plateau as $E_{MS} \rightarrow \infty$ (see Fig. S1 in Supp. Mat.) . This is what is expected in seismology.

In the acoustic experiments reported here, $\beta \approx 1$ (or smaller) and there is an exponential cutoff in $P(E)$ (all the more important since $\beta \approx 1$, or smaller, and, hence its presence is mandatory to define a finite mean value for E). The obtained Bath law is then expected to be different and, in particular, exhibits the unexpected decrease of the magnitude difference with E_{MS} . This comes from the specific form of $P(E)$ here.

This point is now precised in the supplementary materials (pp. 7).

Reviewers' comments:

Reviewer #1 (Remarks to the Author):

The authors have carefully considered the points I raised and I think the rationale and the context for the experimental work and results are now clearer. There is one weakness I think in terms of presentation. "The implications are discussed" is uninformative. It should be deleted or some of the important implications listed explicitly here.

Reviewer #2 (Remarks to the Author):

To summarize the situation from my viewpoint two other reports and mine were provided for this manuscript. The two other referees liked the results and the work a lot whereas my report was the most negative one. Let me now provide with some further commentary on first the authors' response (to my report) and then on the manuscript in its current state.

To summarize, prior to that, my personal – and professional – opinion is that the manuscript may be eventually published if the authors respond to the remaining comments reasonably.

Previous discussion:

1. (On self-organization as a concept): "We disagree with the reviewer on this point. There is an organization of the AEs into aftershock sequences obeying scaling laws". Well this is mostly semantics, but I still feel one needed to insist (self-organization means usually an actual mechanism which is not present here, the AE event series produces apparently the scalings due to the nature of the PDFs not due to any correlations). Good.
2. (On whether the Oslo group had done something similar): "To the very best of our knowledge, neither of these results have been reported on the Oslo group...". This is not quite true since there is one paper at least where they do earthquake catalogs and compare AE and optical measurements; the rest of whether the experiments are similar enough is not relevant but the authors are quite right that the Oslo group did not check these scaling laws (in parenthesis I am at a complete loss as to why not).
3. On AE and depinning dynamics at the origins of the observed avalanches: "There is no one-to-one relation between the two objects. In particular, the AE energy is not proportional to the size of the depinning events." The authors then argue this and that. OK, I believe that the avalanche size distributions measured in the earlier work (Bares et al. PRL) and here are different. This may be due to the details of the AE measuring system which seems to have a very straightforward and simplified definition of energy (energy = peak voltage squared not the integral of the squared signal over the event duration) for instance and/or the list of provided by the authors as suggestions for the difference.
4. (On spatiotemporal correlations): "The spatial localization accuracy of the AE sources is not sufficient to permit such an analysis..." I think here we have a confusion which is partly my fault. My intention was to ask if the AE timeseries has temporal correlations. Are there any?
5. (On the AE event rate): "We do not say that $R(t)$ [the number of AE produced per unit time] is constant..." Here again the point I wanted to say is that in my opinion it would be interesting to see a distribution of $R(t)$ (SI?) as defined over some timescale that makes sense for these experiments.

6. This related to point 4 from above. I return to this issue (depinning transition as a source for the observed scaling) below.

7. OK (minor)

8. OK (minor)

9. OK (minor)

10. (About whether AE energy integral equals crack length). Just a note to the authors. It is somewhat contradictory that the number of events scales with the length better than the energy integral since (in depinning) one would really expect the opposite. The reason is simple: IF I am at the critical point then the crack jumps are power-law distributed (in the forward direction) and in the sum the largest jump dominates unless I have enough jumps to see the cut-off well. I do not think that is the case here with the jump statistics at hand.

On the current manuscript:

1. On p. 3 (paragraph starting "These laws...") the 3rd sentence sounds too general if all lab fracture experiments are intended (minor point). The same paragraph has the potential reason for controversy in that I'd be surprised if the question of the last sentence would not have a "yes" answer in the Oslo results just referred to before. Maybe it would be an idea to suggest testing the scaling laws also in the planar crack case somewhere in the manuscript (conclusions)?

2. Figure 1a has only a snapshot of the actual experiment (in duration). This should be mentioned.

3. Fig. 2B and 2C: given the $P(E)$ I am surprised that one has more than 3000 events with an energy higher than 10^1 . Is this ok?

4. In the results section (2nd and 3rd paragraphs) the authors say that the number of AE events equals the number of heterogeneities met in the direction of propagation by the front. I think this needs an extra comment since the claim is not obvious (though the C 's found are close to this suggestion): one could expect that the typical avalanche sweeps over several of these (as E is power-law distributed). The last sentence of the 1st paragraph in the section says the same thing, but is again in contradiction with avalanches (AE events) being collective events as the $P(E)$ indicates.

5. It would be good to mention explicitly the typical number of events per sample AND the number of samples studied for each disorder.

6. Paragraph "Beyond Gutenberg..." the sentence "For time evolving $R(t)$..." I think should be rather "For widely distributed $R(t)$...". After all, even in this experiment $R(t)$ evolves with time (read: fluctuates around a constant mean).

7. Discussion: The statement "Over the past years..." finishes with a claim that I do not believe. One can drive cracks sub-critical, in creep, or towards the critical point (stress) without any self-adjustment. The idea here is presumably what is called the constant velocity ensemble. The next sentence some people would not believe to be generally true, "AE and depinning events are distinct objects". It may be true in this particular experiment, but I am not really convinced about the reasons even here so as to be certain of what this difference implies.

8. Next, the authors provide four reasons why they see a scale-free distribution of waiting times. The

first of these I think is a wrong explanation. Why should this produce the $P(t)$ measured here? I think the theoretical studies of models of non-local avalanches are clear on this and show if anything the opposite. Point 2 I think the authors could check. There must be – in the timeseries of events – some trace of this due to the correlations it would lead to and checking this is not difficult. Point 3 I do not believe in since I do not understand it as it is written. Taken at face value, the authors seem to imply that a set of subsequent AE events is altogether a consequence of one depinning avalanche. What is the difference of that to point 4 suggested here (where the effect arises from thresholding) for instance?. I do not agree that testing some of these ideas is a significant challenge. It may be so if the AE apparatus of the authors has left them with only the coarse-grained data of event energies and occurrence times (from which the waiting times are presumably calculated as the difference, neglecting the event durations). Still one may study the correlations of the waiting times. Of my comments in this report this is the one most important one to be clear on this.

9. In the next paragraph the authors then note the rough validity of the relation $\alpha = \beta - 1$ in a number of compressional fracture experiments. I agree that this is intriguing, but I not simply understand the sentence "This yields us to argue...". Please elaborate.

10. I have two problems with the final paragraph. First of all, it almost looks as if now the authors would say that look; we can establish all these laws in one depinning problem, so this is a quite general result. Sure, but what about the Omori law here, the power-law waiting times? Or the fact that the authors seem to say that the $P(E)$ measured is something else than what one should measure really here as avalanche statistics "as in a depinning problem". Second, unrelated to that, I fail to see how these results would help to limit crackling noise. In these systems there is one way to do that: stay away from the critical point which means usually making the cut-off of the avalanche (size) distribution as small as possible somehow, to avoid large events. Maybe I miss something in the reasoning of the authors.

Reviewer #3 (Remarks to the Author):

I have checked the revised manuscript and the answers to reviewers. The authors have answered and processed the problem raised by me. Even though the used mathematics are not so rigorous, but they are acceptable.

The only new point is equation (1), it is not the same as the G-R law. The G-R law corresponds to a purely inverse power (Pareto) distribution, while (1) is a tapered Pareto distribution (see Kagan and Schoenberg, 2001. Estimation of the upper cutoff parameter for the tapered Pareto distribution. *J. Appl. Prob.* 38A, 158-175, or Zhuang et al. 2015, Features of the earthquake source process simulated by Vere-Jones' branching crack model. *Bulletin of the Seismological Society of America*. 106, 1832-1839). The tapered tail might be caused by the limitation of the volume of rock material.

I have no other comments and recommend acceptance after very minor revision.

Response to reviewer 1

We would like to thank reviewer 1 again for his/her valuable comments and questions. The identified weakness in terms of presentation has been corrected and the sentence ‘The implication are discussed’ at the end of the introduction has been deleted.

Response to reviewer 3

We would like to thank reviewer 3 again for his/her valuable comments and questions. The new point raised has been addressed and it is now detailed that Equation 1 involves an upper cutoff parameter E_0 (tapered Pareto distribution), while the Gutenberg-Richter law in seismology corresponds to a purely inverse power law (pure Pareto distribution) (see modification of the footnote 1 pp. 6).

The finite system size and/or the limitation of the volume of material might indeed be a relevant parameter. Still, it should be noted that changing the material microstructure (diameter d of the beads prior sintering) while keeping the system size and the material volume significantly modifies the upper cutoff value and the form of the cutoff function (supplementary Fig. S8). The way the acoustic waves attenuate as they propagate within the material (which depends on d) might then be an important parameter to consider. These elements of discussion are now provided (pp. 11, second paragraph in the section “effect of crack speed”).

Response to reviewer 2

We would like to thank reviewer 2 again for her/his valuable comments and questions. We have modified our manuscript accordingly. We hope that this new version properly addresses all the concerns raised by the examiner and can now be published.

In the following, we begin by answering the questions raised following our responses to the series of questions raised in the first round. In the presentation, we chose to keep track of past exchanges to avoid a point being taken out of context: Reviewer comments are in *blue italic* and our response are in black.

In a second step, we respond in details to the new points raised after the first revision.

Previous discussion

POINT 1 -- First of all, I am missing the idea of the "self-organization" the authors make a point about. I think that they rather argue (energies of AE events in particular or the waiting time distribution of events) that there is not much organization at all. I think this is a misleading way of presenting the results.

We disagree with the reviewer on this point. There is an organization of the AEs into aftershock sequences obeying scaling laws reminiscent of those observed in seismology: Productivity law & Eq. 3, Bath law, Omori laws & Eq. 4. Neither of these laws will be fulfilled for non-organized events, i.e. a Poisson process of successive events with « standard » (i.e. not-scale free) energy distribution. This specific organization is demonstrated in our manuscript to be a consequence of the scale-free distribution of both AE energy and interevent times.

For this reason, it is not misleading to speak about the organization of the AEs (even if in our case, the organization does not come from correlations between time occurrence and energy or correlation between the energy of successive events but comes from the scale free distribution of energy and waiting time as we have demonstrated). In the initial version of the manuscript, we call it a « self-organization ». In the revised version, we drop the « self-« which can be misleading (possible confusion between self-organization and self-criticality) : « self-organize » has been replaced by « get organized » and « self-organization » has been replaced by « organization » (abstract pp. 1 & 2).

“We disagree with the reviewer on this point. There is an organization of the AEs into aftershock sequences obeying scaling laws”. Well this is mostly semantics, but I still feel one needed to insist (self-organization means usually an actual mechanism which is not present here, the AE event series produces apparently the scalings due to the nature of the PDFs not due to any correlations). Good.

OK: « self-organize » has been replaced by « get organized » and « self-organization » has been replaced by « organization »

POINT 2 -- The second is that the Oslo group planar crack experiments have demonstrated a large number of these results in a similar setting (in-plane single crack vs. this case where out-of-plane crack excursions also take place). This puts, without the in-depth commentary the manuscript is missing, the novelty of some of the conclusions at doubt.

We also disagree with the reviewer on this point. It is worth to recall here the two main outcomes of our work:

1. The fact that, in a so situation of nominally brittle fracture driven by the propagation of a single crack, AE organize into mainshock-aftershock events obeying the most common seismic laws : Omori-Utsu law (aftershock frequency decays algebraically with time from mainshock), productivity law (number of produced aftershocks increases as a power-law with mainshock energy), and Bath law (difference in magnitude between mainshock and its largest aftershock is independent of the mainshock magnitude).
2. The fact that the above seismic laws for the AS organization directly result from the scale-free statistics of energy (for productivity law and Bath law) and from that of inter-event time (for Omori law), and are not due to time-energy correlations (or spatiotemporal correlations). This is demonstrated in the present manuscript and permit to predict some interrelations between the parameters at play in the seismic laws and that of the scale-free statistics.

To the very best of our knowledge, neither of these results have been reported on the Oslo group planar crack experiments.

It is also worth mentioning that the two setting (ours and Oslo one) are not similar. At least three differences of significant importance have to be mentioned: The dimensionality (2D planar crack in their case vs 3D fracture in ours), the nature of the loading (mixed mode in Oslo case, due to the asymmetry of the loading vs pure mode I in our case) and the studied observables (depinning avalanches in Oslo case vs. acoustic events in ours, see also response to points 3 & 6).

“To the very best of our knowledge, neither of these results have been reported on the Oslo group...”. This is not quite true since there is one paper at least where they do earthquake catalogs and compare AE and optical measurements; the rest of whether the experiments are similar enough is not relevant but the authors are quite right that the Oslo group did not check these scaling laws (in parenthesis I am at a complete loss as to why not).

We do not pretend to be the first to mention statistical similarities between fracture events in experiments at lab scale and the analysis of earthquake catalogs in seismology -- many groups including the one in Oslo have indeed pointed out this similarity. We do say that the two main outcomes of our work are novel and cannot be found in any of the Oslo work (and in any other work). These two main outcomes are recalled to be:

1. In an experiment involving the growth of a single crack under pure tension (nominally brittle crack), AE organize *into mainshock-aftershock events obeying Omori-Utsu law, productivity law and Bath law*;
2. These Omori-Utsu law, productivity law and Bath law are theoretically demonstrated to result from the *individual* power-law distribution of energy and inter-event times, *without requiring the presence of further time-energy correlations*.

POINT 3 -- The third is that this kind of dynamical crack propagation problems are supposed to derive their scaling from the depinning of cracks as elastic manifolds in a random medium - as the Osloc case has shown convincingly in all aspects. Here, while the scale-free distribution of the AE energies follows a power-law as expected, the waiting times do so as well, as is not expected. The authors do note this, in the discussion, and provide three candidate reasons for this discrepancy. I think that this work can not be published without actually explaining this or trying to do so. In particular there are ways of even testing these three ideas (ref. 26 for instance discusses similar issues already!).

It is important to emphasize the differences between AE and depinning avalanches (see also response to point 6). They are not the same objects:

- Depinning avalanches are elastostatic quantities. They correspond to the events when the crack line depins from a heterogeneity and, by doing so, releases part of the elastic (elastostatic) energy stored in the sample by creating a given amount of fracture surface (in the standard depinning framework, the avalanche size is actually defined by this depinning area);
- AE are elastodynamics quantities: they are the signature of the elastic waves triggered by the local accelerations/decelerations going along with the above depinning events? These elastic waves are (locally) collected by the transducers placed on the sample.

There is no one-to-one relation between the two objects. In particular, the AE energy is not proportional to the size of the depinning events. In this context, the discussion section of the first version of the manuscript was misleading. It has been significantly rewritten to emphasize this point (pp. 13).

The fact that the paradigm of depinning elastic manifolds would rather yield to Poissonian waiting time is briefly noted in the discussion of the manuscript, the candidate reasons in its adaptation to the fracture problem that may turn the waiting time distribution to a power-law also. Still, this is clearly not the focus of this manuscript (see response to point 2 which recalls the two main outcomes of our work).

“There is no one-to-one relation between the two objects. In particular, the AE energy is not proportional to the size of the depinning events.” The authors then argue this and that. OK, I believe that the avalanche size distributions measured in the earlier work (Bares et al. PRL) and here are different. This may be due to the details of the AE measuring system which seems to have a very straightforward and simplified definition of energy (energy = peak voltage squared not the integral of the squared signal over the event duration) for instance and/or the list of provided by the authors as suggestions for the difference.

We checked it does not depend on the definition chosen for the AE: The same analysis done after having defined the energy as the integral of the squared signal over the event duration provide similar results. We added a supplementary figure (Fig. S11) showing (i) a plot of the absolute energy (integral of the square signal over the event duration) as a function of the square amplitude peak for all events and (ii) the distribution of energy when it is defined as the integral of the square signal. Energy of AE and of depinning events are truly different for the reasons provided in the first response.

POINT 4 -- One would also think that the authors could probe more deeply into the possible spatiotemporal correlations in the AE timeseries (some such should exist, since the waiting time distribution is not Poissonian).

The spatial localization accuracy of the AE sources is not sufficient to permit such an analysis. This spatial accuracy δx is set by the main frequency of the AE waveform, $40\text{kHz} \leq f \leq 130\text{kHz}$ and the wave speed in the material, $c_W = 2048\text{m/s}$ (for the artificial rock made of $583\mu\text{m}$ PS beads used in the experiments having led to Figs 2 and 3) : $\delta x \sim f/c_W \sim 5\text{ mm}$ to be compared with the specimen thickness of 1.5cm.

The spatial localization accuracy and the way it can be estimated are now provided in the revised manuscript (pp. 17 & 18, section « Methods »).

(On spatiotemporal correlations): “The spatial localization accuracy of the AE sources is not sufficient to permit such an analysis...” I think here we have a confusion which is partly my fault. My intention was to ask if the AE timeseries has temporal correlations. Are there any?

To check this, we used the procedure proposed in Ref. 34 (Stojanova et al. PRL 2014) and computed the repartition of events in 2D energy-waiting time diagrams. For the latter, we considered the waiting time preceding the considered event, Δt_{before} and the one following it, Δt_{after} . As in ref. 34, we computed the number B of events (resp. number A) falling into a box $(\Delta t_{before}, E)$ (resp. in a box $(\Delta t_{after}, E)$) and plotted the resulting 2D maps $B(\Delta t, E)$ and $A(\Delta t, E)$ (see supplementary Figs. S9A and S9B). These maps were compared with a situation where Δt and E are uncorrelated, which was obtained by redistributing randomly the waiting time to be associated with each energy (see supplementary Figs. S9C and S9D). The observation of these relative maps reveals:

- (1) When preceding events are considered, events are observed to concentrate around $(\Delta t_{before}, E) = (0.06\text{s}, E_{min})$ where E_{min} is the lower cutoff for energy (sensitivity of the system) and 0.06s coincide with the characteristic time τ_{min} intervening in the Omori law.
- (2) When following events are considered, there exists a gap at high E/low Δt_{after} . To be more precise, this gap occurs for $\Delta t_{after} < \Delta t_{after}^c \sim E^a$ with $a \sim 1/6$. This is interpreted as the consequence of a *short-time aftershock incompleteness* (STAI), i.e. missing events right after an energetical event – their waveform having been drown in that of the MS (see the point 5 raised by reviewer 3 and our response)
- (3) A (slightly) larger density for small energy and large waiting times (for both preceding and following events) is observed. This indicates the existence of “inactivity times” characterized by long waiting times and low energy events.

The above analysis and associated figures are now provided in supplementary Fig. S9.

POINT 5 -- The authors argue, that the rate R is constant, possibly so, but the timeseries would be interesting anyways

We do not say that $R(t)$ [the number of AE produced per unit time] is constant. In our experiment, we demonstrate that it is the number of AE produced per unit length which is constant (see also response to point 10). As a result, $R(t)$ is proportional to the instantaneous velocity $v(t)$ (with a proportionality constant $\sim H/d^2$ where H is the specimen thickness and d the microstructure length scale, see text), and the mean value \bar{R} is proportional to the mean value \bar{v} over the considered time window.

In the revised manuscript, a figure showing the cumulative number of AE as a function of time has been added (Inset in Fig. 1D). A figure showing the variation of \bar{R} with \bar{v} has also been added in the supplementary materials (Fig. S4).

(On the AE event rate): “We do not say that $R(t)$ [the number of AE produced per unit time] is constant...” Here again the point I wanted to say is that in my opinion it would be interesting to see a distribution of $R(t)$ (SI?) as defined over some timescale that makes sense for these experiments.

This is done in the new version (see supplementary Fig. S5). The obtained distribution compares very well with that observed for the instantaneous velocity (reported in Ref. 27, Bares et al. PRL 2014): The distribution exhibits two scaling regimes: a small scale scaling regime with an exponent ~ 1.4 and a large-scale scaling regime with an exponent ~ 2.5 . This supports our statement $R \sim v$.

POINT 6 -- Is the beta-exponent of $P(E)$ known from earlier work? Or explained by crack depinning?

This is not the case. Here, E is the energy of an acoustic event as it arrives at the piezoelectric transducer, defined as the square of the maximum value $V^2(t)$ of the preamplified voltage $V(t)$ measured on the piezo-acoustic transducer over the duration of the event [It was checked that defining the energy as the integral of $V^2(t)$ over the duration of the event does not change the value of beta]. This acoustic energy is very different from the total elastic energy released by a depinning event, *i.e.* when the crack line locally depins and progresses over one unit: It is orders of magnitude smaller. More importantly, this acoustic energy has no reason to be proportional to the total elastic energy released by a depinning event : the waveform associated with the acoustic pulse will depend on the depinning event, but also on the complete geometry of the specimen at the time of the event, the eigenmodes at that times, their spatial distribution and how they are perceived at the location of the transducer...

Actually, we did measure in an earlier work the distribution of the energy released by the depinning events in the same experiments; the results of the study were reported in Ref. 27. In this case, the energy release was shown to be proportional to the area swept by the depinning avalanches and, as such, has been compared to the predictions of the depinning theory of elastic manifolds. The exponent β' to be associated with this true energy is found:

- To be significantly larger than that measured for the acoustic energy : $\beta' \sim 1.4$ for $v = 2.7 \mu\text{m/s}$, see Ref. 27 Fig. 3A (empty symbols associated with $V_{\text{wedge}} = 16 \text{nm/s}$) to be compared with $\beta \sim 0.96$ for the acoustic energy (fig. 2A of the present manuscript);

- To significantly depend on the mean crack speed, much more than β : $\beta' \sim 1.4$ for $v=2.7\mu\text{m/s}$ ($V_{\text{wedge}} = 16\text{nm/s}$ empty symbols in Fig. 3A of Ref. 27) and $\beta' \sim 1.1$ for $v=27\mu\text{m/s}$ ($V_{\text{wedge}} = 16\text{nm/s}$) filled symbols in fig. 3A Ref. 27) Fig. 3A, filled symbols. As $\beta \sim 0.96$ in the first case, and $\beta \sim 0.93$ in the second case.

The fact that the acoustic energy measured here is very different from the energy released by a depinning event is now emphasized in the manuscript (« Discussion » pp. 13). The fact that the exponent β for the acoustic energy is different from that of true (elastostatic) energy released by the depinning events is also specified (« Discussion pp. 13).

This related to point 4 from above. I return to this issue (depinning transition as a source for the observed scaling) below.

The referee probably means point 3, not point 4. See the first and second response to point 3.

POINT 7 -- The physical meaning of Equation 3 would merit a comment.

The derivation of Equation 3 is based on the fact that the event energies are independent with respect to each others and not correlated with the time occurrence. The derivation is detailed page 5 in supplementary materials. The total number of events with an energy smaller than the prescribed energy E_{MS} for mainshock gives the total number of aftershock (AS) in the catalog (summed over all AS sequences). The total number of event with an energy larger than E_{MS} gives, by definition, the total number of mainshock (MS), and hence the total number of AS sequences. The ration between the two, hence, gives the mean number of AS per sequence which is, by definition, $N_{AS}(E_{MS})$.

The associated text pp. 5 in supplementary materials has been reformulated to make it clearer.

OK (minor)

POINT 8 -- Figure 2b misses unit (time).

This has been corrected.

OK (minor)

POINT 9 -- What is the real time accuracy of the AE system? Milliseconds, microseconds?

The acquisition rate is 40MSample/s. It should also be noted that the minimal interval between two successive events is $402\mu\text{s}$. This duration breaks down into two parts :

- Hit Definition time (HDT) of $400\mu\text{s}$, which is the minimal interval during which the signal should not exceed the threshold after an event initiation to end it ;
- Hit Lockout time (HLT) of $2\mu\text{s}$, which is the interval during which the system remains deaf after the HDT to avoid multiple detections of the same event due to reflexions. In our case, the latter has been reduced to the minimum value available in our system (i.e. $2\mu\text{s}$) due to the small size of our sample.

These information are now provided in the revised manuscript (« methods » section, pp. 17 & 18).

OK (minor)

POINT 10 -- Can one translate the integral of AE energy into crack length? Ie. why should the event number be a good quantity to integrate? Or, better than that integral.

The integral of AE energy cannot be translated into crack length because of the arguments developed in response to point 6. In our experiments, we do observe that the cumulative number of events increase linearly with the crack length, irrespectively of the crack speed. This is an experimental observation. Such observation can be interpreted by stating that the number of AE produced as the crack propagates over a unit length is given by the number of heterogeneities met over this period. Then, the cumulative number of heterogeneities linearly grows with crack length, and so does the cumulative number of produced AE (see Fig. 1D).

If now we plot the integral of the AE energy as a function of crack length (instead of the integral of the number of AEs), we do not observe a straight line (see figure below). This shows that the AE energy is not the good quantity to integrate.

Cumulative AE energy as a function of crack length for $\{d = 583\mu\text{m}, \bar{v} = 2.7\mu\text{m/s}\}$

(About whether AE energy integral equals crack length). Just a note to the authors. It is somewhat contradictory that the number of events scales with the length better than the energy integral since (in depinning) one would really expect the opposite. The reason is simple: IF I am at the critical point then the crack jumps are power-law distributed (in the forward direction) and in the sum the largest jump dominates unless I have enough jumps to see the cut-off well. I do not think that is the case here with the jump statistics at hand.

We agree with this point: For depinning events, the released energy (not the cumulative number of events) will scale with the crack length. Actually, this has been demonstrated experimentally in our previous paper [Ref. 27, Bares et al. PRL 2014] where a special attention was paid to make sure the experimentally measured energy is the total potential energy released by the depinning event. But here, we are monitoring the energy of acoustic events, not that of depinning events and the two are not identical [see point 3].

On the current manuscript:

1. On p. 3 (paragraph starting “These laws..:”) the 3rd sentence sounds too general if all lab fracture experiments are intended (minor point). The same paragraph has the potential reason for controversy in that I’d be surprised if the question of the last sentence would not have a “yes” answer in the Oslo results just referred to before. Maybe it would be an idea to suggest testing the scaling laws also in the planar crack case somewhere in the manuscript (conclusions)?

All lab scale quasi-brittle fracture experiments are intended in the third sentence. We dropped the parenthesis to make it clear.

An outcome of our manuscript is indeed to answer “yes”, Omori-Utsu, productivity and Bath law should be fulfilled in the Oslo experiments. We can also predict the values of the productivity exponent and of the Bath magnitude difference: since $\beta \sim 1.6 - 1.7$ (see Refs. 24 and 26: Maloy et al. PRL 2006, Grob et al. PAG 2009), α is anticipated to be $\alpha \sim 0.6 - 0.7$ (supplementary Eq. S8) and $\Delta M_{\infty} \sim 0.7 - 0.8$ (supplementary Fig. S1B). The suggestion has been added in the conclusion.

2. Figure 1a has only a snapshot of the actual experiment (in duration). This should be mentioned.

This is now mentioned.

3. Fig. 2B and 2C: given the $P(E)$ I am surprised that one has more than 3000 events with an energy higher than 10^1 . Is this ok?

This is OK. The distribution in Figure 2A involves 33481 events. Among them, there are 3543 events of energy larger or equal to 10^1 . The number of Δt between events of energy larger or equal to 10^1 is 8 events less, equal to 3535 events, because the events were collected at 8 different channels and the inter-event time was computed channel-by-channel.

A table has been added in the section “Materials & methods” to provide, for each experiment, the disorder length-scale (the bead diameter prior sintering), the total number of events, the mean number of events per second and, when measured, the mean crack speed (see also point 5).

4. In the results section (2nd and 3rd paragraphs) the authors say that the number of AE events equals the number of heterogeneities met in the direction of propagation by the front. I think this needs an extra comment since the claim is not obvious (though the C 's found are close to this suggestion): one could expect that the typical avalanche sweeps over several of these (as E is power-law distributed). The last sentence of the 1st paragraph in the section says the same thing, but is again in contradiction with avalanches (AE events) being collective events as the $P(E)$ indicates.

Figure 1d and supplementary figure S4 show that the mean number of AE events produced as the crack propagates over a unit length is given by the mean number of heterogeneities met during this propagation. The word mean was missing and has been added.

Nothing *a priori* prevents the crack front to sweep over several heterogeneities certain AE events. Still, this is very unlikely: As mentioned in the section “effect of crack speed”, the duration of the largest AE is measured to be ~ 0.05 s. For the experiment presented in figure 1,

the mean crack speed is $\bar{v} = 2.7\mu\text{m}/\text{s}$. This means that the front typically moves over a distance of ~ 135 nm during the AE of largest durations! Of course, the mean crack speed is not representative of the typical crack speed during a depinning events, which is much larger. Still, Ref. 27 (Bares et al. PRL 2014) permits to estimate an order of magnitude for these depinning speed: $v \sim 1 - 10\text{mm}/\text{s}$ (see Fig. 2C of Ref. 27). This means that the front typically moves over a distance of $\sim 50 - 500\mu\text{m}$ during the AE of largest durations! This displacement is to be compared to the typical heterogeneity size: $d = 583\mu\text{m}$.

This item joins the discussion following items 3 and 6 of the previous discussion. The comparison of the time-scales above provides an additional argument showing that AE events and depinning avalanches are not the same objects. The apparent contradiction mentioned by the reviewer disappears once this point is accepted.

5. It would be good to mention explicitly the typical number of events per sample AND the number of samples studied for each disorder.

This is done in the new version. There were seven samples studied for $d=583\mu\text{m}$ (one per given mean crack speed in Fig. 4A, 4B and 4C). In addition, there were two different samples for $d=24\mu\text{m}$ and two different ones for $d = 233\mu\text{m}$. These additional four experiments were used to make supplementary Fig. S8A and S8B.

A table is now provided in the section “Materials & methods”. It gives, for each of these experiments, the disorder length-scale (the bead diameter prior sintering), the total number of events, the mean number of events per second and the mean crack speed (when measured).

6. Paragraph “Beyond Gutenberg...” the sentence “For time evolving $R(t)$...” I think should be rather “For widely distributed $R(t)$...”. After all, even in this experiment $R(t)$ evolves with time (read: fluctuates around a constant mean).

By “time-evolving $R(t)$...” we mean a nonstationary $R(t)$, i.e. the presence of a trend (defined as a slowly varying component of the time series).

The sentence has been rephrased to make it clearer (see pp. 6 new manuscript).

7. Discussion: The statement “Over the past years...” finishes with a claim that I do not believe. One can drive cracks sub-critical, in creep, or towards the critical point (stress) without any self-adjustment. The idea here is presumably what is called the constant velocity ensemble. The next sentence some people would not believe to be generally true, “AE and depinning events are distinct objects”. It may be true in this particular experiment, but I am not really convinced about the reasons even here so as to be certain of what this difference implies.

To prevent any potential controversy, we dropped the end of the sentence. It now reads: “Over the past years, the avalanche dynamics or crackling noise exhibited by a tensile crack propagating in a heterogeneous solid has found a formulation in terms of a critical depinning transition of a long-range elastic manifold in a random potential”

In the same vein, the sentence “AE and depinning events are distinct objects” is now replaced by “Here, AE and depinning events are distinct objects”

8. Next, the authors provide four reasons why they see a scale-free distribution of waiting times. The first of these I think is a wrong explanation. Why should this produce the $P(t)$ measured here? I think the theoretical studies of models of non-local avalanches are clear on this and show if anything the opposite. Point 2 I think the authors could check. There must be – in the timeseries of events – some trace of this due to the correlations it would lead to and checking this is not difficult. Point 3 I do not believe in since I do not understand it as it is written. Taken at face value, the authors seem to imply that a set of subsequent AE events is altogether a consequence of one depinning avalanche. What is the difference of that to point 4 suggested here (where the effect arises from thresholding) for instance?. I do not agree that testing some of these ideas is a significant challenge. It may be so if the AE apparatus of the authors has left them with only the coarse-grained data of event energies and occurrence times (from which the waiting times are presumably calculated as the difference, neglecting the event durations). Still one may study the correlations of the waiting times. Of my comments in this report this is the one most important one to be clear on this

We reply successively to the various points raised in this comment.

Point 1 (each global depinning avalanche made of several isolated local clusters) was suggested in Ref. 43 (Laurson et al. PRE 2010). It is written in this paper (pp. 046116-6 last paragraph of the paper just before the “Acknowledgment section”) “*The connection between (global) avalanches and (local) clusters opens up also to interesting possibilities to understand... and the foreshock and aftershock sequences of earthquakes*”. We thought it was interesting to make reference to this here [even if the reviewer is quite true in the fact that Ref. 43 reports a simple suggestion, not a demonstration]. On the other hand, this point is rather anecdotal in our present manuscript. Following the suggestion of the reviewer, we dropped it in the revised version.

Point 2 (self-adjustment of the driving force as the mechanism for time organization) cannot be tested with the data we currently have. We do observe some time-correlations in the AE time series (see response to point 4 in the previous discussion and supplementary Fig. S9). However, it should be recalled that, here, AE are not depinning events (see point 3 and our response in the previous discussion). Hence, the eventual correlations ruling the time occurrence of successive depinning events are likely different from those of AE. We follow the reviewer suggestion and delete Point 2 in the new version to avoid potential controversy.

Point 3 was poorly formulated in the previous version. We do it better in the new version (pp. 14). The idea is indeed that a single depinning avalanche leads to a set of several AE as a single “true” depinning avalanche leads to a set of several apparent depinning avalanches when those are defined as the parts above a finite threshold. The difference is that the AE are not directly the emerged parts (i.e. above the threshold) of the depinning avalanche. More likely, they form at the different points of high acceleration/deceleration encountered during the considered avalanche.

The statement “*testing some of these ideas is a significant challenge*” has been deleted.

A characterization of the time correlation of the AE events is now provided and briefly discussed in the supplementary material (supplementary Fig. S9).

9. In the next paragraph the authors then note the rough validity of the relation $\alpha = \beta - 1$ in a number of compressional fracture experiments. I agree that this is intriguing, but I not simply understand the sentence “*This yields us to argue...*”. Please elaborate.

The relations obtained here between the different seismic laws, including the relation $\alpha = \beta - 1$, presuppose the time series are stationary. This assumption is satisfied in our experiment which involves the propagation of a single individual crack.

Damage spreading in compressional quasi-brittle fracture involves two mechanisms:

- Nucleation of microcracks,
- Their individual growth

Continuum damage mechanics puts more emphasis on the first mechanism; the collective, stress-mediated nucleation of the microcracks, their further localization into one or few localization bands is thought to govern the specific organization of the events, and subsequently that of the AE. But this mechanism is clearly not stationary -- the activity rate is low at the beginning and diverges at the overall breakdown of the structure. Hence, it is intriguing to observe $\alpha = \beta - 1$ if the organization of AE in the mentioned compressional experiments is actually governed by a non-stationary mechanism like the collective nucleation of microcracks.

On the other hand, the growth of individual microcracks has no reason to be particularly non-stationary. The fact that our predictions -- recalled to be justified provided a stationary time series -- are fulfilled in these compressive fracture experiments leads us to think that, perhaps, it is rather the second mechanism (the individual growth of the microcracks) than the first one (the collective nucleation of the microcracks) which governs the organization of acoustic events.

The text of the revised manuscript has been amended to make it clearer and less speculative. We now simply note that the inter-relations between the seismic laws unraveled here in a single crack propagation situation and for a stationary time series seem to remain valid in the much more complex situations of compressional fracture, which in particular involve collective nucleation numerous microcracks and clearly non-stationary time series.

10. I have two problems with the final paragraph. First of all, it almost looks as if now the authors would say that look; we can establish all these laws in one depinning problem, so this is a quite general result. Sure, but what about the Omori law here, the power-law waiting times? Or the fact that the authors seem to say that the $P(E)$ measured is something else than what one should measure really here as avalanche statistics "as in a depinning problem". Second, unrelated to that, I fail to see how these results would help to limit crackling noise. In these systems there is one way to do that: stay away from the critical point which means usually making the cut-off of the avalanche (size) distribution as small as possible somehow, to avoid large events. Maybe I miss something in the reasoning of the authors.

The reviewer is true here. The last paragraph has been suppressed.

Reviewers' comments:

Reviewer #2 (Remarks to the Author):

This is my third report on the manuscript, and I think that it has now been substantially improved with all the additions and changes and extra work the authors have put into it. Thus I think I am now in the position of being able to recommend publication after some minor corrections listed below. I also apologize for the delay of my report due to some travel.

Conclusions: "Note however that, here, AE and depinning events are distinct objects, without one-to-one relation between them."

I think I get the point, but the language (English) is to me strange. "distinct objects" is not very understandable.

The authors have provided a new figure S5 about the event rates $R(t)$ with various definitions. I feel a bit confused about what this shows, since it seems to imply there are two power-law regimes for the rates (and the authors seem to imply that this is in agreement somehow with their earlier PRL on depinning in this system). I think the figure caption and/or the main text should explain this better.

Reviewer #3 (Remarks to the Author):

This is the 3rd time I review this article. The author has made the change as I suggested. But I think the authors have made some mistakes on the taper Pareto distribution for earthquake magnitudes. First, E_0 is not the cut-off magnitude, but called the corner magnitude. A cut-off magnitude means that no events above or below it can be observed. In seismological observation, we can observe earthquakes bigger than the corner magnitude, but the frequency of occurrences of such events decays much quicker with magnitude than smaller earthquakes. Second, Line 4 from bottom in Page 11, such a taper Pareto distribution has been observed in seismology. "not observed in seismology" can be deleted.

Response to reviewer 2

We would like to thank reviewer 2 for pointing out these two elements that required to be clarified. Those are addressed below.

Conclusions: "Note however that, here, AE and depinning events are distinct objects, without one-to-one relation between them." I think I get the point, but the language (English) is to me strange. "distinct objects" is not very understandable.

The language has been corrected. The sentence is now replaced by "Note, however, that there is no one-to-one relationship between the depinning events defined above and the acoustic events analyzed here."

The authors have provided a new figure S5 about the event rates $R(t)$ with various definitions. I feel a bit confused about what this shows, since it seems to imply there are two power-law regimes for the rates (and the authors seem to imply that this is in agreement somehow with their earlier PRL on depinning in this system). I think the figure caption and/or the main text should explain this better.

The figure caption has been modified to make it better. The new caption is:

Figure S5 shows the distribution of the "instantaneous" event rate. Note that, in experiments, an "instantaneous" quantity is actually averaged over a finite time scale δt whose value affects the fluctuation amplitude. The distribution of $R(t)$ was therefore computed for different δt (values indicated in the legend). The figure shows that all curves collapse onto a single master curve, exhibiting two power-law regimes: A small scale regime with a scaling exponent $a_{small} = 1.48 \pm 0.16$ and a large scale regime with a scaling exponent $a_{large} = 2.5 \pm 0.3$. Since the mean activity rate is proportional to the mean crack speed (supplementary Fig. S4), it is interesting to compare the above distributions for the event rate, $R(t)$, with the distributions of the "instantaneous" crack speed, $v(t)$ (averaged over the same time scales δt). These have been analyzed in an earlier work [Ref. 27, Bares et al. PRL 2014]. It has been shown [Ref. 27] that, as for $R(t)$, the distributions of $v(t)$ collapse onto a single master curve independent of δt , which exhibits two power-law regimes with similar exponents: $a_{small} = 1.4 \pm 0.15$ and $a_{large} = 2.5 \pm 0.1$. This suggests that the expression "mean activity rate proportional to the mean crack speed" remains true even when averaging is performed over a finite (and relatively small) time scale δt .

Response to reviewer 3

We would like to thank reviewer 3 for pointing out the additional two mistakes in the manuscript that should be corrected. Those are addressed below.

First, E_0 is not the cut-off magnitude, but called the corner magnitude. A cut-off magnitude means that no events above or below it can be observed. In seismological observation, we can observe earthquakes bigger than the corner magnitude, but the frequency of occurrences of such events decays much quicker with magnitude than smaller earthquakes.

We now call E_0 the upper corner energy rather than the cut-off energy [Note that E_0 is an energy in our case, not a magnitude which would be proportional to the logarithmic of E_0]. For the same reasons, we no longer use the terms "upper cutoff" and "lower cutoff" when we discuss the relevant time scales

in the Omori-Utsu law or in the distribution of waiting times. We replaced them by "upper time scale" and "lower time scale".

Second, Line 4 from bottom in Page 11, such a taper Pareto distribution has been observed in seismology. "not observed in seismology" can be deleted.

"not observed in seismology" is now deleted.

Response to reviewer 2

We would like to thank reviewer 2 again for her/his valuable comments and questions. We have modified our manuscript accordingly. We hope that this new version addresses properly all the concerns raised by the examiner and can now be published.

In the following, we begin by answering the questions raised following our responses to the series of questions raised in the first round; we took care to trace all the exchanges in order to avoid a point being taken out of context.

In a second step, we respond in details to the new points raised after the first revision.

Previous discussion

POINT 1 -- First of all, I am missing the idea of the "self-organization" the authors make a point about. I think that they rather argue (energies of AE events in particular or the waiting time distribution of events) that there is not much organization at all. I think this is a misleading way of presenting the results.

We disagree with the reviewer on this point. There is an organization of the AEs into aftershock sequences obeying scaling laws reminiscent of those observed in seismology: Productivity law & Eq. 3, Bath law, Omori laws & Eq. 4. Neither of these laws will be fulfilled for non-organized events, i.e. a Poisson process of successive events with « standard » (i.e. not-scale free) energy distribution. This specific organization is demonstrated in our manuscript to be a consequence of the scale-free distribution of both AE energy and interevent times.

For this reason, it is not misleading to speak about the organization of the AEs (even if in our case, the organization does not come from correlations between time occurrence and energy or correlation between the energy of successive events but comes from the scale free distribution of energy and waiting time as we have demonstrated). In the initial version of the manuscript, we call it a « self-organization ». In the revised version, we drop the « self-« which can be misleading (possible confusion between self-organization and self-criticality) : « self-organize » has been replaced by « get organized » and « self-organization » has been replaced by « organization » (abstract pp. 1 & 2).

“We disagree with the reviewer on this point. There is an organization of the AEs into aftershock sequences obeying scaling laws”. Well this is mostly semantics, but I still feel one needed to insist (self-organization means usually an actual mechanism which is not present here, the AE event series produces apparently the scalings due to the nature of the PDFs not due to any correlations). Good.

OK: « self-organize » has been replaced by « get organized » and « self-organization » has been replaced by « organization »

POINT 2 -- The second is that the Oslo group planar crack experiments have demonstrated a large number of these results in a similar setting (in-plane single crack vs. this case where out-of-plane crack excursions also take place). This puts, without the in-depth commentary the manuscript is missing, the novelty of some of the conclusions at doubt.

We also disagree with the reviewer on this point. It is worth to recall here the two main outcomes of our work:

1. The fact that, in a so situation of nominally brittle fracture driven by the propagation of a single crack, AE organize into mainshock-aftershock events obeying the most common seismic laws : Omori-Utsu law (aftershock frequency decays algebraically with time from mainshock), productivity law (number of produced aftershocks increases as a power-law with mainshock energy), and Bath law (difference in magnitude between mainshock and its largest aftershock is independent of the mainshock magnitude).
2. The fact that the above seismic laws for the AS organization directly result from the scale-free statistics of energy (for productivity law and Bath law) and from that of inter-event time (for Omori law), and are not due to time-energy correlations (or spatiotemporal correlations). This is demonstrated in the present manuscript and permit to predict some interrelations between the parameters at play in the seismic laws and that of the scale-free statistics.

To the very best of our knowledge, neither of these results have been reported on the Oslo group planar crack experiments.

It is also worth mentioning that the two setting (ours and Oslo one) are not similar. At least three differences of significant importance have to be mentioned: The dimensionality (2D planar crack in their case vs 3D fracture in ours), the nature of the loading (mixed mode in Oslo case, due to the asymmetry of the loading vs pure mode I in our case) and the studied observables (depinning avalanches in Oslo case vs. acoustic events in ours, see also response to points 3 & 6).

“To the very best of our knowledge, neither of these results have been reported on the Oslo group...”. This is not quite true since there is one paper at least where they do earthquake catalogs and compare AE and optical measurements; the rest of whether the experiments are similar enough is not relevant but the authors are quite right that the Oslo group did not check these scaling laws (in parenthesis I am at a complete loss as to why not).

We do not pretend to be the first to mention statistical similarities between fracture events in experiments at lab scale and the analysis of earthquake catalogs in seismology -- many groups including the one in Oslo have indeed pointed out this similarity. We do say that the two main outcomes of our work are novel and cannot be found in any of the Oslo work (and in any other work). These two main outcomes are recalled to be:

1. In an experiment involving the growth of a single crack under pure tension (nominally brittle crack), AE organize into mainshock-aftershock events obeying Omori-Utsu law, productivity law and Bath law;
2. These Omori-Utsu law, productivity law and Bath law are theoretically demonstrated to result from the individual power-law distribution of energy and inter-event times, without requiring the presence of further time-energy correlations.

POINT 3 -- The third is that this kind of dynamical crack propagation problems are supposed to derive their scaling from the depinning of cracks as elastic manifolds in a random medium - as the Osloc case has shown convincingly in all aspects. Here, while the scale-free distribution of the AE energies follows a power-law as expected, the waiting times do so as well, as is not expected. The authors do note this, in the discussion, and provide three candidate reasons for this discrepancy. I think that this work can not be published without actually explaining this or trying to do so. In particular there are ways of even testing these three ideas (ref. 26 for instance discusses similar issues already!).

It is important to emphasize the differences between AE and depinning avalanches (see also response to point 6). They are not the same objects:

- Depinning avalanches are elastostatic quantities. They correspond to the events when the crack line depins from a heterogeneity and, by doing so, releases part of the elastic (elastostatic) energy stored in the sample by creating a given amount of fracture surface (in the standard depinning framework, the avalanche size is actually defined by this depinning area);
- AE are elastodynamics quantities: they are the signature of the elastic waves triggered by the local accelerations/decelerations going along with the above depinning events? These elastic waves are (locally) collected by the transducers placed on the sample.

There is no one-to-one relation between the two objects. In particular, the AE energy is not proportional to the size of the depinning events. In this context, the discussion section of the first version of the manuscript was misleading. It has been significantly rewritten to emphasize this point (pp. 13).

The fact that the paradigm of depinning elastic manifolds would rather yield to Poissonian waiting time is briefly noted in the discussion of the manuscript, the candidate reasons in its adaptation to the fracture problem that may turn the waiting time distribution to a power-law also. Still, this is clearly not the focus of this manuscript (see response to point 2 which recalls the two main outcomes of our work).

“There is no one-to-one relation between the two objects. In particular, the AE energy is not proportional to the size of the depinning events.” The authors then argue this and that. OK, I believe that the avalanche size distributions measured in the earlier work (Bares et al. PRL) and here are different. This may be due to the details of the AE measuring system which seems to have a very straightforward and simplified definition of energy (energy = peak voltage squared not the integral of the squared signal over the event duration) for instance and/or the list of provided by the authors as suggestions for the difference.

We checked it does not depend on the definition chosen for the AE: The same analysis done after having defined the energy as the integral of the squared signal over the event duration provide similar results. We added a supplementary figure (Fig. S11) showing 1) a plot of the absolute energy (integral of the square signal over the event duration) as a function of the square amplitude peak for all events and 2) the distribution of energy when it is defined as the integral of the square signal. Energy of AE and of depinning events are truly different for the reasons provided in the first response.

POINT 4 -- One would also think that the authors could probe more deeply into the possible spatiotemporal correlations in the AE timeseries (some such should exist, since the waiting time distribution is not Poissonian).

The spatial localization accuracy of the AE sources is not sufficient to permit such an analysis. This spatial accuracy δx is set by the main frequency of the AE waveform, $40\text{kHz} \leq f \leq 130\text{kHz}$ and the wave speed in the material, $c_W = 2048\text{m/s}$ (for the artificial rock made of $583\mu\text{m}$ PS beads used in the experiments having led to Figs 2 and 3) : $\delta x \sim f/c_W \sim 5\text{ mm}$ to be compared with the specimen thickness of 1.5cm.

The spatial localization accuracy and the way it can be estimated are now provided in the revised manuscript (pp. 17 & 18, section « Methods »).

(On spatiotemporal correlations): “The spatial localization accuracy of the AE sources is not sufficient to permit such an analysis...” I think here we have a confusion which is partly my fault. My intention was to ask if the AE timeseries has temporal correlations. Are there any?

To check this, we used the procedure proposed in Ref. 34 (Stojanova et al. PRL 2014) and computed the repartition of events in 2D energy-waiting time diagrams. For the latter, we considered the waiting time preceding the considered event, Δt_{before} and the one following it, Δt_{after} . As in ref. 34, we computed the number B of events (resp. number A) falling into a box $(\Delta t_{\text{before}}, E)$ (resp. in a box $(\Delta t_{\text{after}}, E)$) and plotted the resulting 2D maps $B(\Delta t, E)$ and $A(\Delta t, E)$ (see supplementary Figs. S9A and S9B). These maps were compared with a situation where Δt and E are uncorrelated, which was obtained by redistributing randomly the waiting time to be associated with each energy (see supplementary Figs. S9C and S9D). The observation of these relative maps reveals:

- (1) When preceding events are considered, events are observed to concentrate around $(\Delta t_{\text{before}}, E) = (0.06\text{s}, E_{\text{min}})$ where E_{min} is the lower cutoff for energy (sensitivity of the system) and 0.06s coincide with the characteristic time τ_{min} intervening in the Omori law.
- (2) When following events are considered, there exists a gap at high E/low Δt_{after} . To be more precise, this gap occurs for $\Delta t_{\text{after}} < \Delta t_{\text{after}}^c \sim E^a$ with $a \sim 1/6$. This is interpreted as the consequence of a *short-time aftershock incompleteness* (STAI), i.e. missing events right after an energetical event – their waveform having been drown in that of the MS (see the point 5 raised by reviewer 3 and our response)
- (3) A (slightly) larger density for small energy and large waiting times (for both preceding and following events) is observed. This indicates the existence of “inactivity times” characterized by long waiting times and low energy events.

The above analysis and associated figures are now provided in supplementary Fig. S9.

POINT 5 -- The authors argue, that the rate R is constant, possibly so, but the timeseries would be interesting anyways

We do not say that $R(t)$ [the number of AE produced per unit time] is constant. In our experiment, we demonstrate that it is the number of AE produced per unit length which is constant (see also response to point 10). As a result, $R(t)$ is proportional to the instantaneous velocity $v(t)$ (with a proportionality constant $\sim H/d^2$ where H is the specimen thickness and d the microstructure length scale, see text), and the mean value \bar{R} is proportional to the mean value \bar{v} over the considered time window.

In the revised manuscript, a figure showing the cumulative number of AE as a function of time has been added (Inset in Fig. 1D). A figure showing the variation of \bar{R} with \bar{v} has also been added in the supplementary materials (Fig. S4).

(On the AE event rate): “We do not say that $R(t)$ [the number of AE produced per unit time] is constant...” Here again the point I wanted to say is that in my opinion it would be interesting to see a distribution of $R(t)$ (SI?) as defined over some timescale that makes sense for these experiments.

This is done in the new version (see supplementary Fig. S5). The obtained distribution compares very well with that observed for the instantaneous velocity (reported in Ref. 27, Bares et al. PRL 2014): The distribution exhibits two scaling regimes: a small scale scaling regime with an exponent ~ 1.4 and a large-scale scaling regime with an exponent ~ 2.5 . This supports our statement $R \sim v$.

POINT 6 -- Is the beta-exponent of $P(E)$ known from earlier work? Or explained by crack depinning?

This is not the case. Here, E is the energy of an acoustic event as it arrives at the piezoelectric transducer, defined as the square of the maximum value $V^2(t)$ of the preamplified voltage $V(t)$ measured on the piezo-acoustic transducer over the duration of the event [It was checked that defining the energy as the integral of $V^2(t)$ over the duration of the event does not change the value of beta]. This acoustic energy is very different from the total elastic energy released by a depinning event, *i.e.* when the crack line locally depins and progresses over one unit: It is orders of magnitude smaller. More importantly, this acoustic energy has no reason to be proportional to the total elastic energy released by a depinning event : the waveform associated with the acoustic pulse will depend on the depinning event, but also on the complete geometry of the specimen at the time of the event, the eigenmodes at that times, their spatial distribution and how they are perceived at the location of the transducer...

Actually, we did measure in an earlier work the distribution of the energy released by the depinning events in the same experiments; the results of the study were reported in Ref. 27. In this case, the energy release was shown to be proportional to the area swept by the depinning avalanches and, as such, has been compared to the predictions of the depinning theory of elastic manifolds. The exponent β' to be associated with this true energy is found:

- To be significantly larger than that measured for the acoustic energy : $\beta' \sim 1.4$ for $v = 2.7 \mu\text{m/s}$, see Ref. 27 Fig. 3A (empty symbols associated with $V_{\text{wedge}} = 16 \text{nm/s}$) to be compared with $\beta \sim 0.96$ for the acoustic energy (fig. 2A of the present manuscript);

- To significantly depend on the mean crack speed, much more than β : $\beta' \sim 1.4$ for $v=2.7\mu\text{m/s}$ ($V_{\text{wedge}} = 16\text{nm/s}$ empty symbols in Fig. 3A of Ref. 27) and $\beta' \sim 1.1$ for $v=27\mu\text{m/s}$ ($V_{\text{wedge}} = 16\text{nm/s}$) filled symbols in fig. 3A Ref. 27) Fig. 3A, filled symbols. As $\beta \sim 0.96$ in the first case, and $\beta \sim 0.93$ in the second case.

The fact that the acoustic energy measured here is very different from the energy released by a depinning event is now emphasized in the manuscript (« Discussion » pp. 13). The fact that the exponent β for the acoustic energy is different from that of true (elastostatic) energy released by the depinning events is also specified (« Discussion pp. 13).

This related to point 4 from above. I return to this issue (depinning transition as a source for the observed scaling) below.

The referee probably means point 3, not point 4. See the first and second response to point 3.

POINT 7 -- The physical meaning of Equation 3 would merit a comment.

The derivation of Equation 3 is based on the fact that the event energies are independent with respect to each others and not correlated with the time occurrence. The derivation is detailed page 5 in supplementary materials. The total number of events with an energy smaller than the prescribed energy E_{MS} for mainshock gives the total number of aftershock (AS) in the catalog (summed over all AS sequences). The total number of event with an energy larger than E_{MS} gives, by definition, the total number of mainshock (MS), and hence the total number of AS sequences. The ration between the two, hence, gives the mean number of AS per sequence which is, by definition, $N_{AS}(E_{MS})$.

The associated text pp. 5 in supplementary materials has been reformulated to make it clearer.

OK (minor)

POINT 8 -- Figure 2b misses unit (time).

This has been corrected.

OK (minor)

POINT 9 -- What is the real time accuracy of the AE system? Milliseconds, microseconds?

The acquisition rate is 40MSample/s. It should also be noted that the minimal interval between two successive events is 402 μs . This duration breaks down into two parts :

- Hit Definition time (HDT) of 400 μs , which is the minimal interval during which the signal should not exceed the threshold after an event initiation to end it ;
- Hit Lockout time (HLT) of 2 μs , which is the interval during which the system remains deaf after the HDT to avoid multiple detections of the same event due to reflexions. In our case, the latter has been reduced to the minimum value available in our system (i.e. 2 μs) due to the small size of our sample.

These information are now provided in the revised manuscript (« methods » section, pp. 17 & 18).

OK (minor)

POINT 10 -- Can one translate the integral of AE energy into crack length? Ie. why should the event number be a good quantity to integrate? Or, better than that integral.

The integral of AE energy cannot be translated into crack length because of the arguments developed in response to point 6. In our experiments, we do observe that the cumulative number of events increase linearly with the crack length, irrespectively of the crack speed. This is an experimental observation. Such observation can be interpreted by stating that the number of AE produced as the crack propagates over a unit length is given by the number of heterogeneities met over this period. Then, the cumulative number of heterogeneities linearly grows with crack length, and so does the cumulative number of produced AE (see Fig. 1D).

If now we plot the integral of the AE energy as a function of crack length (instead of the integral of the number of AEs), we do not observe a straight line (see figure below). This shows that the AE energy is not the good quantity to integrate.

Cumulative AE energy as a function of crack length for $\{d = 583\mu\text{m}, \bar{v} = 2.7\mu\text{m/s}\}$

(About whether AE energy integral equals crack length). Just a note to the authors. It is somewhat contradictory that the number of events scales with the length better than the energy integral since (in depinning) one would really expect the opposite. The reason is simple: IF I am at the critical point then the crack jumps are power-law distributed (in the forward direction) and in the sum the largest jump dominates unless I have enough jumps to see the cut-off well. I do not think that is the case here with the jump statistics at hand.

We agree with this point: For depinning events, the released energy (not the cumulative number of events) will scale with the crack length. Actually, this has been demonstrated experimentally in our previous paper [Bares et al. PRL 2014] where a special attention was paid to make sure the experimentally measured energy is the total potential energy released by the depinning event. But here, we are monitoring the energy of acoustic events, not that of depinning events and the two are not identical [see point 3].

1. On p. 3 (paragraph starting “These laws..:”) the 3rd sentence sounds too general if all lab fracture experiments are intended (minor point). The same paragraph has the potential reason for controversy in that I’d be surprised if the question of the last sentence would not have a “yes” answer in the Oslo results just referred to before. Maybe it would be an idea to suggest testing the scaling laws also in the planar crack case somewhere in the manuscript (conclusions)?

All lab scale quasi-brittle fracture experiments are intended in the third sentence. We dropped the parenthesis to make it clear.

An outcome of our manuscript is indeed to answer “yes”, Omori-Utsu, productivity and Bath law should be fulfilled in the Oslo experiments. We can also predict the values of the productivity exponent and of the Bath magnitude difference: since $\beta \sim 1.6 - 1.7$ (see Refs. 24 and 26: Maloy et al. PRL 2006, Grob et al. PAG 2009), α is anticipated to be $\alpha \sim 0.6 - 0.7$ (supplementary Eq. S8) and $\Delta M_{\infty} \sim 0.7 - 0.8$ (supplementary Fig. S1B). The suggestion has been added in the conclusion.

2. Figure 1a has only a snapshot of the actual experiment (in duration). This should be mentioned.

This is now mentioned.

3. Fig. 2B and 2C: given the $P(E)$ I am surprised that one has more than 3000 events with an energy higher than 10^1 . Is this ok?

This is OK. The distribution in Figure 2A involves 33481 events. Among them, there are 3543 events of energy larger or equal to 10^1 . The number of Δt between events of energy larger or equal to 10^1 is 8 events less, equal to 3535 events, because the events were collected at 8 different channels and the inter-event time was computed channel-by-channel.

A table has been added in the section “Materials & methods” to provide, for each experiment, the disorder length-scale (the bead diameter prior sintering), the total number of events, the mean number of events per second and, when measured, the mean crack speed (see also point 5).

4. In the results section (2nd and 3rd paragraphs) the authors say that the number of AE events equals the number of heterogeneities met in the direction of propagation by the front. I think this needs an extra comment since the claim is not obvious (though the C 's found are close to this suggestion): one could expect that the typical avalanche sweeps over several of these (as E is power-law distributed). The last sentence of the 1st paragraph in the section says the same thing, but is again in contradiction with avalanches (AE events) being collective events as the $P(E)$ indicates.

Figure 1d and supplementary figure S4 show that the mean number of AE events produced as the crack propagates over a unit length is given by the mean number of heterogeneities met during this propagation. The word mean was missing and has been added.

Nothing *a priori* prevents the crack front to sweep over several heterogeneities certain AE events. Still, this is very unlikely: As mentioned in the section “effect of crack speed”, the duration of the largest AE is measured to be ~ 0.05 s. For the experiment presented in figure 1,

the mean crack speed is $\bar{v} = 2.7\mu\text{m}/\text{s}$. This means that the front typically moves over a distance of ~ 135 nm during the AE of largest durations! Of course, the mean crack speed is not representative of the typical crack speed during a depinning events, which is much larger. Still, Ref. 27 (Bares et al. PRL 2014) permits to estimate an order of magnitude for these depinning speed: $v \sim 1 - 10\text{mm}/\text{s}$ (see Fig. 2C of Ref. 27). This means that the front typically moves over a distance of $\sim 50 - 500\mu\text{m}$ during the AE of largest durations! This displacement is to be compared to the typical heterogeneity size: $d = 583\mu\text{m}$.

This item joins the discussion following items 3 and 6 of the previous discussion. The comparison of the time-scales above provides an additional argument showing that AE events and depinning avalanches are not the same objects. The apparent contradiction mentioned by the reviewer disappears once this point is accepted.

5. *It would be good to mention explicitly the typical number of events per sample AND the number of samples studied for each disorder.*

This is done in the new version. There were seven samples studied for $d=583\mu\text{m}$ (one per given mean crack speed in Fig. 4A, 4B and 4C). In addition, there were two different samples for $d=24\mu\text{m}$ and two different ones for $d = 233\mu\text{m}$. These additional four experiments were used to make supplementary Fig. S8A and S8B.

A table is now provided in the section “Materials & methods”. It gives, for each of these experiments, the disorder length-scale (the bead diameter prior sintering), the total number of events, the mean number of events per second and the mean crack speed (when measured).

6. *Paragraph “Beyond Gutenberg...” the sentence “For time evolving $R(t)$...” I think should be rather “For widely distributed $R(t)$...”. After all, even in this experiment $R(t)$ evolves with time (read: fluctuates around a constant mean).*

By “time-evolving $R(t)$...” we mean a nonstationary $R(t)$, i.e. the presence of a trend (defined as a slowly varying component of the time series).

The sentence has been rephrased to make it clearer (see pp. 6 new manuscript).

7. *Discussion: The statement “Over the past years....” finishes with a claim that I do not believe. One can drive cracks sub-critical, in creep, or towards the critical point (stress) without any self-adjustment. The idea here is presumably what is called the constant velocity ensemble. The next sentence some people would not believe to be generally true, “AE and depinning events are distinct objects”. It may be true in this particular experiment, but I am not really convinced about the reasons even here so as to be certain of what this difference implies.*

To prevent any potential controversy, we dropped the end of the sentence. It now reads: “Over the past years, the avalanche dynamics or crackling noise exhibited by a tensile crack propagating in a heterogeneous solid has found a formulation in terms of a critical depinning transition of a long-range elastic manifold in a random potential”

In the same vein, the sentence “AE and depinning events are distinct objects” is now replaced by “Here, AE and depinning events are distinct objects”

8. Next, the authors provide four reasons why they see a scale-free distribution of waiting times. The first of these I think is a wrong explanation. Why should this produce the $P(t)$ measured here? I think the theoretical studies of models of non-local avalanches are clear on this and show if anything the opposite. Point 2 I think the authors could check. There must be – in the timeseries of events – some trace of this due to the correlations it would lead to and checking this is not difficult. Point 3 I do not believe in since I do not understand it as it is written. Taken at face value, the authors seem to imply that a set of subsequent AE events is altogether a consequence of one depinning avalanche. What is the difference of that to point 4 suggested here (where the effect arises from thresholding) for instance?. I do not agree that testing some of these ideas is a significant challenge. It may be so if the AE apparatus of the authors has left them with only the coarse-grained data of event energies and occurrence times (from which the waiting times are presumably calculated as the difference, neglecting the event durations). Still one may study the correlations of the waiting times. Of my comments in this report this is the one most important one to be clear on this

We reply successively to the various points raised in this comment.

Point 1 (each global depinning avalanche made of several isolated local clusters) was suggested in Ref. 43 (Laurson et al. PRE 2010). It is written in this paper (pp. 046116-6 last paragraph of the paper just before the “Acknowledgment section”) “*The connection between (global) avalanches and (local) clusters opens up also to interesting possibilities to understand... and the foreshock and aftershock sequences of earthquakes*”. We thought it was interesting to make reference to this here [even if the reviewer is quite true in the fact that Ref. 43 reports a simple suggestion, not a demonstration]. On the other hand, this point is rather anecdotal in our present manuscript. Following the suggestion of the reviewer, we dropped it in the revised version.

Point 2 (self-adjustment of the driving force as the mechanism for time organization) cannot be tested with the data we currently have. We do observe some time-correlations in the AE time series (see response to point 4 in the previous discussion and supplementary Fig. S9). However, it should be recalled that, here, AE are not depinning events (see point 3 and our response in the previous discussion). Hence, the eventual correlations ruling the time occurrence of successive depinning events are likely different from those of AE. We follow the reviewer suggestion and delete Point 2 in the new version to avoid potential controversy.

Point 3 was poorly formulated in the previous version. We do it better in the new version (pp. 14). The idea is indeed that a single depinning avalanche leads to a set of several AE as a single “true” depinning avalanche leads to a set of several apparent depinning avalanches when those are defined as the parts above a finite threshold. The difference is that the AE are not directly the emerged parts (i.e. above the threshold) of the depinning avalanche. More likely, they form at the different points of high acceleration/deceleration encountered during the considered avalanche.

The statement “*testing some of these ideas is a significant challenge*” has been deleted.

A characterization of the time correlation of the AE events is now provided and briefly discussed in the supplementary material (supplementary Fig. S9).

9. In the next paragraph the authors then note the rough validity of the relation $\alpha = \beta - 1$ in a number of compressional fracture experiments. I agree that this is intriguing, but I not simply understand the sentence “*This yields us to argue...*”. Please elaborate.

The relations obtained here between the different seismic laws, including the relation $\alpha = \beta - 1$, presuppose the time series are stationary. This assumption is satisfied in our experiment which involves the propagation of a single individual crack.

Damage spreading in compressional quasi-brittle fracture involves two mechanisms:

- Nucleation of microcracks,
- Their individual growth

Continuum damage mechanics puts more emphasis on the first mechanism; the collective, stress-mediated nucleation of the microcracks, their further localization into one or few localization bands is thought to govern the specific organization of the events, and subsequently that of the AE. But this mechanism is clearly not stationary -- the activity rate is low at the beginning and diverges at the overall breakdown of the structure. Hence, it is intriguing to observe $\alpha = \beta - 1$ if the organization of AE in the mentioned compressional experiments is actually governed by a non-stationary mechanism like the collective nucleation of microcracks.

On the other hand, the growth of individual microcracks has no reason to be particularly non-stationary. The fact that our predictions -- recalled to be justified provided a stationary time series -- are fulfilled in these compressive fracture experiments leads us to think that, perhaps, it is rather the second mechanism (the individual growth of the microcracks) than the first one (the collective nucleation of the microcracks) which governs the organization of acoustic events.

The text of the revised manuscript has been amended to make it clearer and less speculative. We now simply note that the inter-relations between the seismic laws unraveled here in a single crack propagation situation and for a stationary time series seem to remain valid in the much more complex situations of compressional fracture, which in particular involve collective nucleation numerous microcracks and clearly non-stationary time series.

10. I have two problems with the final paragraph. First of all, it almost looks as if now the authors would say that look; we can establish all these laws in one depinning problem, so this is a quite general result. Sure, but what about the Omori law here, the power-law waiting times? Or the fact that the authors seem to say that the $P(E)$ measured is something else than what one should measure really here as avalanche statistics "as in a depinning problem". Second, unrelated to that, I fail to see how these results would help to limit crackling noise. In these systems there is one way to do that: stay away from the critical point which means usually making the cut-off of the avalanche (size) distribution as small as possible somehow, to avoid large events. Maybe I miss something in the reasoning of the authors.

The reviewer is true here. The last paragraph has been suppressed.

REVIEWERS' COMMENTS:

Reviewer #2 (Remarks to the Author):

The authors have responded satisfactorily to the last comments, thus I recommend publication.

Reviewer #3 (Remarks to the Author):

The manuscript is acceptable now for publication. I have no further comments or requests for revision .

The referees's comments are included below. They do not raised any further issues.

Reviewer #2: The authors have responded satisfactorily to the last comments, thus I recommend publication.

Reviewer #3: The manuscript is acceptable now for publication. I have no further comments or requests for revision.